

# Observations and modelling of glyoxal in the tropical Atlantic marine boundary layer

Hannah Walker,[1,2] Daniel Stone,[2] Trevor Ingham,[2,3] Sina Hackenberg,[4] Danny Cryer,[2] Shalini Punjabi,[4] Katie Read,[4,5] James Lee,[4,5] Lisa Whalley,[2,3] Dominick V. Spracklen,[1] Lucy J. Carpenter,[4] Steve R. Arnold,[1,*] Dwayne E. Heard[2*]

[1] School of Earth and Environment, University of Leeds, Leeds, LS2 9JT, UK

[2] School of Chemistry, University of Leeds, Leeds, LS2 9JT, UK

[3] National Centre for Atmospheric Science, School of Chemistry, University of Leeds, Leeds, LS2 9JT, UK

[4] Wolfson Atmospheric Chemistry Laboratories, Department of Chemistry, University of York, Heslington, York, YO10 5DD, UK

[5] National Centre for Atmospheric Science, Wolfson Atmospheric Chemistry Laboratories, Department of Chemistry, University of Leeds, Leeds, LS2 9JT, UK

* Corresponding authors: s.arnold@leeds.ac.uk, d.e.heard@leeds.ac.uk

## Abstract

In situ field measurements of glyoxal at the surface in the tropical marine boundary layer have been made with a temporal resolution of a few minutes during two 4-week campaigns in June-July and August-September 2014 at the Cape Verde Atmospheric Observatory (CVAO, 16° 52' N, 24° 52' W). Using laser-induced phosphorescence spectroscopy with an instrumental detection limit of ~ 1 pptv (1 hour averaging), volume mixing ratios up to ~10 pptv were observed, with 24 hour averaged mixing ratios of 4.9 pptv and 6.3 pptv observed during the first and second campaigns, respectively. Some diel behaviour was observed but this was not marked. A box model using the detailed Master Chemical Mechanism (version 3.2) and constrained with detailed observations of a suite of species co-measured at the observatory was used to calculate glyoxal mixing ratios. There is a general model underestimation of the glyoxal observations during both campaigns, with mean midday (1100-1300 hours) observed-to-modelled ratios for glyoxal of 3.2 and 4.2 for the two campaigns, respectively, and higher ratios at night. A rate of production analysis shows the dominant sources of glyoxal in this environment to be the reactions of OH with glycoaldehyde and acetylene, with a significant contribution from the reaction of OH with the peroxide HC(O)CH$_2$OOH, which itself derives from OH oxidation of acetaldehyde. Increased mixing ratios of acetaldehyde, which is unconstrained and potentially underestimated in the base model, can significantly improve the agreement between the observed and modelled glyoxal during the day. Mean midday observed-to-modelled glyoxal ratios decreased to 1.3 and 1.8 for campaigns 1 and 2, respectively, on constraint to a fixed acetaldehyde mixing ratio of 200 pptv, which is consistent with recent airborne measurements near CVAO. However, a significant model underprediction remains at night. The model was sensitive to changes in deposition rates of model intermediates and the uptake of glyoxal onto aerosol. The midday (1100-1300) mean modelled glyoxal mixing ratio decreased by factors of 0.87 and 0.90 on doubling the deposition rates of model intermediates and aerosol uptake of glyoxal, respectively, and increased by factors of 1.10 and 1.06 on halving the deposition rates of model intermediates and aerosol uptake of glyoxal, respectively. Although measured



levels of monoterpenes at the site (total of ~ 1 pptv) do not significantly influence the model calculated levels of glyoxal, transport of air from a source region with high monoterpene emissions to the site has the potential to give elevated mixing ratios of glyoxal from monoterpene oxidation products, but the values are highly sensitive to the deposition rates of these oxidised intermediates. A source of glyoxal derived from production in the ocean surface organic microlayer cannot be ruled out on the basis of this work, and may be significant at night.

## 1. Introduction

Reactive organic compounds in the marine atmosphere have the potential to influence climate through changing the atmospheric oxidation capacity (Read et al., 2012; Whalley et al., 2010) and by modifying the composition and size distribution of remote aerosol (O'Dowd et al., 2004; Sracklen et al., 2008). Oxidation by the hydroxyl radical (OH) in the tropical marine lower troposphere accounts for around 20-25% of the global atmospheric sink of the greenhouse gas methane (Bloss et al., 2005). Perturbations to the OH abundance in these regions by marine-sourced reactive organic compounds therefore have the potential to indirectly impact climate by modifying the atmospheric methane lifetime (Read et al., 2012).

In situ observational studies demonstrate a strong correlation between the organic mass fraction of marine aerosol and ocean biological activity (O'Dowd et al., 2004). Satellite observations also suggest a relationship between ocean biology, cloud albedo and cloud droplet number concentration (CDNC) in remote marine regions (Meskhidze & Nenes, 2006; Krüger & Graßl, 2011), implying a biological source for marine cloud condensation nuclei (CCN). The mechanism for this relationship is not known, but a possibility is biologically driven production and subsequent ocean emission of organics, acting directly as aerosol or indirectly as aerosol precursors. Due to the profound influence of marine stratocumulus clouds on global climate, there is a high priority to understand cloud condensation nuclei (CCN) budgets in the remote marine regions. At the low CCN concentrations typical of the marine environment, cloud properties are highly sensitive to changes in CCN and respond non-linearly to aerosol concentrations (Platnick & Twomey, 1994).

Glyoxal (CHOCHO) is the simplest di-carbonyl species, and is highly reactive with an estimated global mean atmospheric lifetime of around 3 hours, with loss mostly driven by photolysis (Fu et al., 2008). Oxidation of glyoxal by reaction with OH or by photolysis leads to the rapid production of peroxy radicals, in particular $HO_2$, and condensable products, which are believed to lead to the formation of secondary organic aerosol (SOA) (Liggio et al., 2005; Fu et al., 2008). Glyoxal has been shown to enhance the growth of nanoparticles, producing non-volatile oligomers in the particle phase (Wang et al., 2010). Marine emission and subsequent oxidation of biogenic volatile organic compounds (VOCs) has been shown to form secondary organic aerosol (SOA) in the marine environment (Knote et al., 2014; Volkamer et al., 2015; Chiu et al., 2017). While marine primary organic aerosol (POA) has received much attention, much less is understood regarding marine SOA sources. At least five separate biologically-driven model source estimates have been produced in an attempt to simulate the oceanic POA source to the atmosphere, and these are largely capable of reproducing observed concentrations of water-insoluble organic matter (WIOM) in the remote



marine boundary layer (MBL) (Gantt et al., 2012). However, models typically underestimate remote marine water-soluble organic carbon aerosol (WSOC) by factors of 3-6 in the North Atlantic and southern Indian Ocean (Meskhidze et al., 2011). Model underestimation of WSOC may point to an underestimation in marine SOA (Ceburnis et al., 2008; Facchini et al., 2008). While oxidation and ageing of primary aerosol organics may also contribute some fraction of WSOC (Ovadnevaite et al., 2011), several precursor VOCs which are known to produce SOA have been observed in the remote marine environment, including isoprene (Yokouchi et al., 1999; Lewis et al., 2001; Matsunaga et al., 2002; Yassaa et al., 2008, Colomb et al., 2009; Hackenberg et al., 2017a; Kim et al., 2017), monoterpenes (Yassaa et al., 2008; Hackenberg et al., 2017b; Kim et al., 2017), and glyoxal (Mahajan et al., 2014; Coburn et al., 2014; Lawson et al., 2015). According to limited observations, WSOC makes up 25-60 % of marine organic aerosol in clean marine air masses (O'Dowd et al., 2004; Meskhidze et al., 2011), so knowledge of its sources is important for understanding aerosol-cloud interactions in the marine environment.

To date, reported measurements of glyoxal concentrations in the remote marine atmosphere have mostly been based on Long-Path Differential Optical Absorption Spectroscopy (LP-DOAS) and Multi-Axis DOAS (MAX-DOAS) techniques. Based on a combined dataset of measurements made using these techniques from 10 field campaigns between 2009 and 2013, average mixing ratios in the open ocean marine boundary layer are ~24 pptv, with an upper limit of 40 pptv (Mahajan et al., 2014). The largest mixing ratios have been measured near coasts, and these measurements show no evidence for enhancement of glyoxal over remote tropical oceans as suggested by some satellite measurements (Vrekoussis et al., 2009; Lerot et al., 2010). In situ observations have been made using the Fast Light-Emitting Diode Cavity-Enhanced Differential Optical Absorption Spectroscopy (Fast LED-CE-DOAS) technique, showing average glyoxal mixing ratios of 43 pptv and 32 pptv in the Tropical Pacific MBL in the northern and southern hemisphere, respectively (Coburn et al., 2014). These surface observations are a similar magnitude to in situ airborne measurements made over the tropical eastern Pacific Ocean using MAX-DOAS, which showed concentrations of 34 ± 7 pptv at 250 m, in good agreement with measurements by the ship-based MAX-DOAS (33 ± 7 pptv at 100 m) and Fast LED-CE-DOAS (38 ± 5 pptv at 18 m) for 30 s integration times (Volkamer et al., 2015). Eddy covariance measurements using the Fast LED-CE-DOAS technique showed a daytime flux in both hemispheres from the atmosphere into the ocean, with ocean to atmosphere fluxes observed during the night in the southern hemisphere and just after sunset in the northern hemisphere, possibly indicative of a source of glyoxal from the ocean surface organic microlayer (Coburn et al., 2014). 2,4-dinitrophenylhydrazine (2,4-DNPH) cartridges (24-hour samples) and high-performance liquid chromatography (HPLC) was used to measure glyoxal concentrations during a cruise over Chatham Rise in the south-west Pacific Ocean in February and March 2012, and from the Cape Grim Baseline Air Pollution Station during August and September 2011 (Lawson et al., 2015). 24-hr average glyoxal mixing ratios observed in clean marine air were 23 ± 8 pptv over Chatham Rise and 7 ± 2 pptv at Cape Grim, substantially lower than previous remote sensing measurements. Comparison with concurrent vertical column densities from GOME-2 satellite measurements revealed that the satellite observations exceeded the in situ observations by more than $1.5 \times 10^{14}$ molecule cm$^{-2}$ at both sites.



There are no studies that have used coincident in situ observations of glyoxal and its precursor species from the remote marine boundary layer to constrain its sources in the marine atmosphere. Typical marine boundary layer mixing ratios of isoprene (10 pptv) and acetylene (200 pptv) cannot account for even the lower glyoxal concentrations, suggesting an unknown source of glyoxal in these regions (Sinreich et al., 2010; Lawson et al., 2015). Ozonolysis of biogenic primary emitted monoterpenes, including $\alpha$-pinene, carene, geraniol, and citral (Yu et al., 1998; Fick et al., 2003; Nunes et al., 2005) is known to be an important glyoxal source (Fu et al., 2008). Although enhanced monoterpene concentrations in the marine atmosphere have been observed (Yassaa et al., 2008; Luo & Yu, 2010), the role of a marine glyoxal source from monoterpene oxidation has not yet been investigated. Direct ocean emission has been shown to be an important source of carbonyl species to the remote marine atmosphere, including acetaldehyde (Millet et al., 2010; Read et al., 2012; Wang et al., 2019) and acetone (Jacob et al., 2002; Fischer et al., 2012; Read et al., 2012; Wang et al., 2020). However, the high solubility of glyoxal and rapid timescales for hydration reactions of glyoxal in the ocean negate the viability of an appreciable sea-air flux of glyoxal as a result of glyoxal production in subsurface ocean waters (Volkamer et al., 2009; Ervens and Volkamer 2010). Despite this, there is evidence to suggest that glyoxal is produced photochemically in the sea surface microlayer (SML) (Zhou and Mopper, 1997, van Pinxteren and Herrmann, 2013; Chiu et al., 2017). Satellite-based measurements of gas-phase glyoxal show that its spatial distribution is similar to that of chlorophyll-$a$, (Fu et al., 2008). Despite the potential for spectral interference in the glyoxal retrieval from ocean colour in regions of biological activity, this spatial relationship may indicate a source of glyoxal which is linked to biological activity in marine regions (Chiu et al., 2017).

Here we report the first high temporal resolution (sub-hourly) in situ observations of glyoxal in the remote marine boundary layer over periods of multiple weeks during two separate field campaigns at the same tropical Atlantic site. We use a sensitive in situ laser-induced phosphorescence (LIP) technique, allowing measurements with a detection limit of ~1 pptv for 1 hour averaging but recorded with a duty cycle of 7 minutes (with a corresponding average limit of detection (LOD) of ~3 pptv). We use these new observations in conjunction with photochemical box modelling and direct observations of glyoxal precursor species, to investigate the dominant processes controlling its formation and loss in the remote marine atmosphere, and the extent to which observed concentrations can be reconciled with our knowledge of sources and sinks. Section 2 provides an overview of the observation site and a description of the new LIP instrument, Section 3 presents the glyoxal measurements, Section 4 describes the model simulations and comparison with observations, and Section 5 evaluates the model performance and implications of our findings for the remote marine glyoxal budget.

## 2. Oceanic Reactive Carbon: Chemistry-Climate Impacts (ORC[3]) campaigns and glyoxal observations

### 2.1. Site overview

Two intensive measurement campaigns were undertaken as part of the Oceanic Reactive Carbon: Chemistry-Climate Impacts (ORC[3]) project at the Global Atmospheric Watch (GAW) Cape Verde Atmospheric Observatory (CVAO) station (16° 52' N, 24° 52' W, http://ncasweb.leeds.ac.uk/capeverde/). The CVAO is positioned on the north-eastern side of São Vicente in the Cape Verde archipelago in the tropical Eastern North Atlantic Ocean, located ~500 miles west of



Senegal, and receives clean, well processed marine air from the north east more than 95% of the time (Carpenter et al., 2010). The site is minimally influenced by local effects and intermittent continental pollution. As a volcanic island, São Vicente is characterised by an absence of coastal features such as extensive shallows or large seaweed beds, which may otherwise provide a source for emissions, for example of halogenated species, particularly at low tide. The tropical MBL has been noted as a key region for glyoxal (Wittrock et al., 2006; Sinreich et al., 2007; Fu et al., 2008; Sinreich et al., 2010; Volkamer et al., 2015; Coburn et al., 2014; Mahajan et al., 2014) and the location of CVAO is thus well suited to this study.

The ORC$^3$ campaigns were each of 4 weeks in duration, and took place from 22$^{nd}$ June to 15$^{th}$ July 2014 and 18$^{th}$ August to 15$^{th}$ September 2014. Between June and October, air masses arriving at the site are typically dominated by marine air from the North Atlantic with an influence from the African coast. Solar radiation tends to peak in May/June, while maximum temperatures and humidity are typically observed in September. Long-term observations at the site since 2006 have indicated the influence of ocean-derived volatile organic compounds during all seasons (Carpenter et al., 2010).

**2.2. Glyoxal measurements**

Observations of glyoxal were performed in situ using an instrument based on laser induced phosphorescence (LIP) spectroscopy, which was custom built for the ORC$^3$ campaign. The methodology is similar to that developed by Keutsch and co-workers (the Madison laser-induced phosphorescence instrument, MADLIP), which has previously demonstrated sensitive field detection of glyoxal with a reported 60 s LOD of 18 pptv (Huismann et al., 2008), and which was then improved to 1 pptv in 5 min (Henry et al., 2012). LIP has previously been used to detect ambient glyoxal in forested, urban, and polluted rural environments (Huisman et al., 2011, DiGangi et al., 2012, Ahlm et al., 2012, Pusede et al., 2014). Here, we make the first deployment of this in situ technique that displays both high spatial and temporal resolution to measure glyoxal in the remote marine boundary layer. The low detection limits are particularly important for measurements of very low ambient concentrations expected in this environment. Prior to the current work, glyoxal has not been measured above an instrument's limit of detection (for example one based on DOAS with a detection limit of ~150 pptv) at the Cape Verde Atmospheric Observatory (Mahajan et al., 2010).

In the LIP technique, glyoxal is excited by laser radiation at $\lambda$ = 440.141 nm from v"=0 of its ground $S_0$ ($^1A_g$) electronic state to $v_8'$=1 of the first excited singlet state $S_1$ ($^1A_u$), where $v_8$ is the C-H bond wagging vibrational mode. Excitation is followed by intersystem crossing to vibrationally excited levels of the first excited triplet state $T_1$ ($^3A_u$) with near-unity yield at pressures exceeding 1 Torr, with subsequent non-radiative relaxation to $v'$ = 0 in the $T_1$ state (Anderson et al., 1973). The transition from the $T_1$ state to the $S_0$ state results in phosphorescence, with the greatest phosphorescence intensity involving the $^3A_u$ ($v_8'$=0) $\rightarrow$ $^1A_g$ ($v_0''$=0) transition at $\lambda$ = 520.8 nm (Holzer and Ramsay, 1970). The phosphorescence collection optics are therefore optimised to transmit this wavelength (details given below). The phosphorescence lifetime, controlled by collisional quenching of the $T_1$ state by $O_2$, was determined in the laboratory to be ~13 μs in 100 Torr of air. Photon counting is therefore delayed until well after the prompt scattered light from



the laser pulse (pulse width ~35 ns) and any short-lived fluorescence inside the cell have completely decayed, thereby

2    minimising the background signal.

4    **2.2.1 Instrument Description**

The main components of the instrument are a phosphorescence cell with an inlet and detector, a laser system, control

6    computer, and a vacuum pump. All components apart from the vacuum pump are built into a double-width 19-inch

aircraft rack.

Laser radiation at $\lambda$ ~ 440 nm is generated by a diode-pumped Nd:YAG-pumped tunable Ti:Sapphire laser (Photonics

10    Industries DS-532-10 and TU-UV-308). The laser wavelength is tuned by changing the angle of the grating in the

Ti:Sapphire cavity with a motorised rotation stage (Newport Corporation URS75BPP precision rotation stage controlled

12    by Newport Corporation SMC100PP single-axis stepper motor controller/driver), giving rapid and reproducible

wavelength changes (absolute accuracy ± 0.015 °; bi-directional repeatability ± 0.01 °; maximum speed 40 ° s$^{-1}$). The

14    Ti:Sapphire laser is tuned to $\lambda$ = 880 nm and frequency-doubled using a lithium borate (LBO) doubling crystal to

produce  30 to 50 mW of light at $\lambda$ ~ 440 nm with a pulse width of ~ 35 ns and a linewidth of ~ 0.043 cm$^{-1}$. The

16    arrangement of the laser, optics, and phosphorescence cell are shown in Figure 1.

18    The laser beam is directed into a 210 mm long baffled input arm by two turning mirrors (1 inch diameter broadband

dielectric coating; reflectivity > 99 % at $\lambda$ = 400 to 750 nm), and is aligned through the centre of the input and output

20    arms. Laser power is measured by a calibrated photodiode (UDT Instruments UDT-555UV) at the end of the 210 mm

long baffled output arm. The scattered light contribution to the background signal is minimised by focussing and

22    collimating the laser beam using two lenses (25 mm diameter, 50 mm focal length, uncoated and 25 mm diameter, 25

mm focal length, uncoated), to give a beam width of approximately 1 mm. An iris immediately before the baffled input

24    arm further reduces stray laser light inside the cell. A small fraction of the laser beam is diverted to a wavemeter

(Coherent Wavemaster; resolution = ± 0.001 nm; accuracy = ± 0.005 nm) using an uncoated glass flat, enabling

26    continuous monitoring of the laser wavelength.

28    The phosphorescence cell is a 110 mm tall, black anodised aluminium cylinder with an internal diameter of 50 mm.

The cylindrical housings for the photodetector and the retroreflector, and the baffled laser light input and output

30    arms, are sealed onto the sides of the cell by round, uncoated Suprasil windows between two o-rings. The detection

assembly and laser beam optics are housed in a custom-built optical enclosure to exclude ambient light from the

32    background signal.

34    Ambient air is drawn through a 1/8 inch internal diameter stainless steel Swagelok union mounted in the top of the

phosphorescence cell, 65 mm above the centre of the phosphorescence region. The fitting facilitates connection to a

36    stainless steel quarter turn instrument plug valve and ¼ inch outer diameter (0.156 inch internal diameter) PFA

sampling line. A dry scroll pump (Agilent Technologies, IDP-3) provides a constant pumping speed of 60 L min$^{-1}$,





reduced to ~ 5 L min$^{-1}$ by a butterfly valve downstream of the phosphorescence cell. The cell is connected to the valve by ¼ inch PFA tubing connected to four ports in the side of the cell below the phosphorescence region. The valve is connected to the pump by 2 m of NW16 flexible steel bellows. The input and output arms are also evacuated by the pump to prevent ambient air recirculating into the phosphorescence region. The optimum cell pressure for the measurement of glyoxal, determined in the laboratory, is a function of the excitation rate of glyoxal and the rate of quenching of phosphorescence by $O_2$. The pressure-dependent LIP signal has a very broad maximum at 100 Torr. We observed fluctuations in cell pressure (± 10 Torr) during the campaign and attributed these to changes in demand from the sampling manifold. The broad pressure dependence means that this did not affect the sensitivity of the instrument.

Glyoxal undergoes reversible wall loss, with equilibrated walls serving as a reservoir for glyoxal (Loza et al., 2010, Kroll et al., 2005). Huisman et al. (2008) found that heating the detection cell to 35 °C was sufficient to prevent loss of glyoxal to the walls of the cell, and this method was adopted here. The sample line is heated using trace heating tape. The inlet and detection cell are heated by two cartridge heaters (RS Components, 20 W, 120 V). The heaters are controlled by two temperature controllers (CAL PID temperature controller 3300) with thermocouples inside and outside the cell and sampling line.

Photons are detected on an axis perpendicular to the laser beam and the gas flow (see Figure 1). Light from the phosphorescence region passes through an interference filter (Semrock FF02-520/28, centre wavelength = 520 nm, > 93 % average transmission, 28 nm minimum bandwidth) to remove laser scattered light and is focussed by two plano-convex lenses (2 inch diameter, 52 mm focal length, anti-reflection coated at $\lambda$ = 521 nm) onto the photocathode of a photomultiplier tube (PMT) (Sens-Tech P25PC photodetector module; 15 % quantum efficiency at $\lambda$ = 520 nm). A retroreflector (CVI Optics fused silica plano-concave spherical mirror, broadband coated, radius of curvature = 50.8 mm) positioned opposite the detector maximises the collection of light from the phosphorescence region. The detector signal is received by a photon counting card (Becker & Hickl, PMS 400) in the data acquisition computer.

Figure 2 shows the timing scheme for signal acquisition. A delay generator (Berkeley Nucleonics Corporation, Model 555 Delay-Pulse Generator) triggers each laser pulse and the photon counting card. The start of each cycle is defined by the laser trigger. The Ti:Sapphire laser pulse occurs 5 μs after the trigger. The laser excites fluorescence from the anodising dye on the cell walls, surfaces of optics, and gas-phase species such as $NO_2$. A delay of 3.0 μs between the laser pulse and the commencement of photon counting ensures that this short-lived fluorescence and any prompt laser scattered light have decayed before the glyoxal phosphorescence signal plus any background signal are recorded in a 35 μs wide integration window (gate A in Figure 2). After a further delay of 50 μs the background signal alone is recorded in a 105 μs wide integration window (gate B in Figure 2). As the phosphorescence is significantly red-shifted (~520 nm) from the laser excitation wavelength (440 nm), switching the PMT off during the laser-pulse to avoid possible saturation or overload is not necessary, and the PMT remains in a high gain state throughout the cycle. The signal is integrated over 1 s, corresponding to 5000 laser pulses, and is recorded by the computer.





The signal collected in gate A is the sum of glyoxal phosphorescence and background signal, which has contributions from PMT dark counts, ambient scattered light, and any laser scattered light which has not completely decayed before photon-counting begins. However, the signal collected in gate B is from the background only but excludes laser-scattered light. The PMT has a specified dark count rate of ~200 s⁻¹, giving a maximum dark count of ~35 s⁻¹ when integrated over the 5000 repetitions of Gate A (35 µs) in one second. The observed background signal, after subtraction of the dark counts, is on the order of 20 s⁻¹ and is dominated by long-lived laser scattered light.

The signal from glyoxal phosphorescence and laser-scattered light is calculated as:

$$Sig_{\text{Gly}} = Sig_{\text{A}} - \frac{Sig_{\text{B}}}{x} \qquad\qquad \text{(Equation 1)}$$

where $Sig_{\text{Gly}}$ is the signal from glyoxal phosphorescence plus laser scattered light, $Sig_{\text{A}}$ is the signal collected in gate A, $Sig_{\text{B}}$ is the signal collected in gate B, and $x = 3$ is the ratio of the width of gate B to the width of gate A. The 1 Hz $Sig_{\text{Gly}}$ data are then normalised by laser power. A measurement duty cycle of 300 seconds online and 120 seconds offline optimises the limit of detection without compromising the applicability of the offline measurement or the stability of the laser wavelength and alignment. To subtract the contribution from laser-scattered light, the mean normalised offline signal is subtracted from the mean normalised online signal for each duty cycle to give the glyoxal signal, $S_{\text{Gly}}$. Figure 3 shows a laser-induced phosphorescence excitation spectrum of glyoxal recorded with the LIP instrument in the laboratory. In ambient measurements, the concentration of glyoxal is determined by tuning the laser on and off the absorption or online wavelength of 440.141 nm. An offline wavelength of 440.035 nm results in an online:offline ratio of the absorption coefficients of more than 3, and switching between the two wavelengths is rapidly and reproducibly achieved by the laser tuning system.

**2.2.2 Instrument Calibration**

The glyoxal signal, $S_{\text{Gly}}$, is related to the ambient glyoxal mixing ratio by the sensitivity or calibration factor, $C$.

$$[\text{Glyoxal}]/\text{pptv} = \frac{S_{\text{Gly}}}{C} \qquad\qquad \text{(Equation 2)}$$

A known mixing ratio of glyoxal is produced by the reaction of OH with acetylene and is used to determine $C$. A flow of synthetic air controlled by a mass flow controller (Brooks 5850S) is humidified by diverting a variable portion through an ambient temperature distilled water bubbler. The humidified flow is directed down a 30 cm long square-section aluminium flow tube (1.27 cm internal dimension) at ~ 50 slm. A mercury pen-ray lamp (Oriel Instruments 6035) is fixed side-on to the flow tube and a quartz (Suprasil) window transmits light from the lamp at λ =184.9 nm. Photolysis of the water vapour produces a known concentration of OH, which is determined by actinometric calibration of the lamp using $N_2O$ photolysis to give NO (Edwards et al., 2003). A mass flow controller (Brooks 5850S) delivers 450 sccm of dilute acetylene (BOC, custom mix of 2 % of acetylene in $N_2$) which is injected into the flow downstream of the lamp. Complete conversion of OH occurs in the flow before it is sampled by the instrument. The glyoxal concentration is given by:





$$[\text{Gly}] = 1.75 \cdot [\text{OH}] = 1.75 \cdot [\text{H}_2\text{O}] \cdot \sigma_{\text{H}_2\text{O}} \cdot \phi_{\text{OH}} \cdot Ft \qquad \text{(Equation 3)}$$

2  where $\sigma_{\text{H}_2\text{O}}$ is the absorption cross section of water vapour at λ = 184.9 nm (7.14 ± 0.2 × $10^{-20}$ molecule$^{-1}$ cm$^{-2}$)

(Cantrell et al., 1997), $\phi_{\text{OH}}$ is the quantum yield of OH at λ = 184.9 nm, which is equal to unity, and $Ft$ is the product

4  of the lamp flux and the photolysis time, determined by actinometry. The yield of glyoxal from OH + acetylene (1.75)

under the conditions of the calibration was determined by modelling the reaction in the chemical kinetics program

6  Kintecus (Ianni, 2017), using a reaction mechanism and kinetics data from the Master Chemical Mechanism

(http://mcm.leeds.ac.uk/MCM/home.htt; Jenkin et al., 2003; Saunders et al., 2003) and Lockhart et al. (2013). The

8  larger than unity factor of 1.75 between [Gly] and [OH] in equation (3) is a result of the OH initiated oxidation of

acetylene in air giving the product HC(O-O)=C(OH)H, which isomerises to form HC(OOH)-C(O)H, which subsequently

10  decomposes to eliminate OH and forming glyoxal, with the OH going on to react further with acetylene to form

additional glyoxal.

A range of glyoxal mixing ratios can be achieved during a calibration by changing the lamp flux. Data are processed as

14  described above and are averaged over each mixing ratio. A straight line, weighted by the standard deviations of the

mixing ratio of glyoxal and the LIP signal is fitted to the averaged data, with the sensitivity or calibration factor, $C$,

16  being equal to the gradient. The uncertainty in the calibration factor is calculated as the sum in quadrature of the

standard error of the gradient and the combined uncertainty in glyoxal yield, [H$_2$O], $\sigma_{\text{H}_2\text{O}}$, $Ft$ , online laser

18  wavelength and laser power. Figure 4 shows a typical calibration obtained during the ORC$^3$ campaign, with the

calibration factor ranging between 8.25 ± 2.21 × $10^{-3}$ count s$^{-1}$ mW$^{-1}$ pptv$^{-1}$ and 1.17 ± 0.34 × $10^{-2}$ count s$^{-1}$ mW$^{-1}$

20  pptv$^{-1}$ during the ORC$^3$ fieldwork.

22  **2.2.3 Limit of Detection**

The minimum detectable glyoxal mixing ratio of the instrument is limited by the signal-to-noise ratio ($SNR$), which

24  depends on the sensitivity and the background signal, and for Poisson statistics:

$$[\text{Gly}]_{\text{min}} = \frac{SNR}{C \times P} \sigma_{\text{offline}} \sqrt{\frac{1}{m} + \frac{1}{n}} \qquad \text{(Equation 4)}$$

26  where $C$ is the sensitivity or calibration factor, $P$ is the online laser power, $\sigma_{\text{offline}}$ is the standard deviation of the

unnormalised offline signal (typically ~ 6 count s$^{-1}$), $m$ is the number of 1s online data points, n is the number of 1s

28  offline data points

30  Since the instrument was deployed in the field for the first time during ORC$^3$, the measurement duty cycle was initially

set at 60 s online and 60 s offline ($m$=$n$=60) to minimise any instabilities in the background signal, the laser wavelength

32  or alignment caused by operation of the instrument under non-laboratory conditions. The limit of detection during

this phase ranged from 5.98 to 11.06 pptv. Both the online and offline measurement periods were extended later in

34  the campaign once the stability of the instrument was characterised, to minimise the limit of detection, which then





ranged from 2.55 pptv to 9.42 pptv for an acquisition cycle of 300 s online ($m=300$) and 120 s ($n=120$) offline. For comparison with model data, the glyoxal measurements were averaged to one hour, giving limits of detection between 0.76 pptv and 6.94 pptv calculated using $m=2571$ and $n=1029$ in Equation 4.

### 2.2.4 Determination of the instrument zero

During ORC[3] fieldwork the zero of the instrument was determined by periodically sampling synthetic air (BOC BTCA 178) and recording measurements over a number of duty cycles (each consisting of 300 s online and 120 s offline). Any difference between the mean online and offline normalised signals was then taken as the instrument zero offset. The zero was determined several times during the campaign, and the mean offset was 0.013 count $s^{-1}$ $mW^{-1}$, or 1.29 pptv, with a standard deviation of 0.039 count $s^{-1}$ $mW^{-1}$ or 3.83 pptv. Ambient concentrations were obtained by subtracting the relevant zero value from $S_{Gly}$ before the calibration factor was applied.

### 2.2.5 Measurement uncertainty

The measurement uncertainty, $\sigma_{Gly}$, for short averaging times is determined by the uncertainties in the calibration factor and the instrument zero which are given by the standard error in the linear fit to the calibration data and the standard deviation of the zero measurements, respectively. The mean $1\sigma$ uncertainty for averaging times up to 7 minutes was 74 % during the first ORC[3] measurement campaign, and 59 % during the second campaign. For 1 hour averaged data the $1\sigma$ uncertainties are 27 % for the first campaign and 26 % for the second campaign.

### 2.3 Supporting measurements

A suite of supporting measurements were made during both campaigns, including mixing ratios of $O_3$, CO, $NO_x$ (NO, $NO_2$), $NO_y$, $C_2$ to $C_8$ non-methane volatile organic compounds (NMVOCs), $j(O^1D)$, wavelength-dependent incoming solar radiation, wind speed and direction, air pressure and temperature, and relative humidity. Details of supporting measurements used to examine the glyoxal observations reported in this work made during ORC[3] are summarised in Table 1, with time series shown in the Supporting Information (Figures S1 and S2). Further details of the measurement techniques, calibration, and accuracy, are given in previous work (Lee et al., 2010; Carpenter et al., 2010).





| | Mean (all data) | Median (all data) | Mean (campaign 1) | Median (campaign 1) | Mean (campaign 2) | Median (campaign 2) |
|---|---|---|---|---|---|---|
| Pressure / hPa | 1013 ± 0 | 1013 | 1013 ± 0 | 1013 | 1013 ± 0 | 1013 |
| Temperature / K | 298 ± 1 | 298 | 297 ± 1 | 297 | 299 ± 1 | 299 |
| Water vapour / ppm | 26414 ± 2812 | 26136 | 24126 ± 1244 | 24114 | 28815 ± 1825 | 28794 |
| $O_3$ / ppb | 25 ± 5 | 25 | 28 ± 5 | 28 | 23 ± 4 | 22 |
| $NO_2$ / pptv | 21 ± 13 | 18 | 25 ± 11 | 22 | 17 ± 15 | 12 |
| NO / pptv | 4 ± 36 | 1 | 2 ± 3 | 1 | 10 ± 70 | 1 |
| CO / pptv | 80 ± 7 | 80 | 79 ± 5 | 79 | 81 ± 8 | 82 |
| $C_2H_6$ / pptv | 454 ± 170 | 480 | 487 ± 195 | 511 | 426 ± 125 | 433 |
| $C_3H_8$ / pptv | 17 ± 11 | 17 | 19 ± 14 | 17 | 16 ± 8 | 17 |
| $IC_4H_{10}$ / pptv | 1 ± 2 | 0 | 1 ± 1 | 0 | 2 ± 2 | 2 |
| $NC_4H_{10}$ / pptv | 3 ± 3 | 2 | 2 ± 2 | 2 | 3 ± 3 | 3 |
| $C_5H_8$ / pptv | 1 ± 1 | 0 | 1 ± 1 | 1 | 0 ± 1 | 0 |
| $IC_5H_{12}$ / pptv | 1 ± 3 | 0 | 1 ± 3 | 0 | 1 ± 3 | 0 |
| $NC_5H_{12}$ / pptv | 0 ± 1 | 0 | 0 ± 1 | 0 | 0 ± 1 | 0 |
| $NC_6H_{14}$ / pptv | 9 ± 4 | 9 | 10 ± 5 | 10 | 8 ± 3 | 8 |
| $C_2H_4$ / pptv | 31 ± 12 | 32 | 28 ± 11 | 30 | 34 ± 11 | 35 |
| $C_3H_6$ / pptv | 14 ± 6 | 15 | 12 ± 6 | 13 | 16 ± 6 | 17 |
| Benzene / pptv | 6 ± 3 | 6 | 6 ± 4 | 6 | 7 ± 3 | 7 |
| Toluene / pptv | 1 ± 1 | 0 | 1 ± 2 | 0 | 0 ± 1 | 0 |
| $C_2H_2$ / pptv | 34 ± 23 | 39 | 33 ± 24 | 37 | 35 ± 22 | 40 |

2    Table 1: Summary of supporting measurements at the Cape Verde Atmospheric Observatory during the ORC[3] campaign for periods during which glyoxal measurements were made. Zero values indicate measurements below the

4    instrumental limit of detection. Chemical names are those used in the MCM. Details of measurement techniques, integration times and uncertainties are given by Carpenter et al. (2010).

**2.4 Deployment of the instrument at Cape Verde**

8    The glyoxal instrument was housed in a 40 ft long air-conditioned shipping container which has been converted into two laboratories, located on the north (windward) side of the site, approximately 55 m from the coastline. Ambient

10    air was sampled 10 m above the ground (20 m above mean sea level) from a tower on the roof of the container into a 40 mm internal diameter glass tube (the sampling manifold), heated to 40 °C to minimise the loss of analytes. The flow

12    rate through the sampling manifold, which was used to sample glyoxal, $NO_x$, $O_3$, CO, and VOCs was maintained at 50 L $min^{-1}$ by a KNF diaphragm vacuum pump (N 035.1.2 AT18 with an IP 44 motor) located between the two laboratories

14    inside the container. The glyoxal inlet was connected to a glass tee piece in the sampling manifold by 5 m of ¼ inch outer diameter PFA (perfluoroalkoxy alkane) tubing (wall thickness 0.047 inch; internal diameter 0.156 inch), which

16    was heated to 40 °C and insulated. The residence time between the start of sampling manifold (main inlet) and the glyoxal fluorescence region was approximately 30 seconds. In addition to using the sampling manifold, on two

18    consecutive days during the campaign sampling was also performed for short periods through a short length of unheated tube directly out of the side of the container, with no noticeable difference observed in measured glyoxal

20    concentrations.



### 3. Overview of glyoxal measurements during ORC[3]

The instrument was operational for 24 days and 22 nights during the first campaign, and for 25 days and 21 nights during the second campaign. Wave breaking activity on the shore close to the measurement site produces a high concentration of sea spray aerosol. Particles that reached the detection cell of the glyoxal instrument and overlapped with the laser beam caused a sharp increase in laser scattered light and were detected as an easily noticed large spike in the data. These spikes were removed by excluding any single 1 Hz data point that exceeded 5 times the standard deviation above the mean value over 1 duty cycle. After filtering there were 250 hours (358 hours in total) of online measurements from the first campaign, and 227 hours (318 hours in total) from the second campaign.

Figure 5 shows the time series of glyoxal observations throughout both campaigns, and demonstrates that glyoxal was above the LOD throughout most of the ORC[3] campaign. The maximum observed mixing ratio during both campaigns was 36.3 pptv, measured on 22nd June 2014, although the 24 hour mean mixing ratio during campaign 1 was lower than that during campaign 2, with values of 4.9 pptv and 6.3 pptv, respectively. Figure S3 summarises the distributions of observed glyoxal mixing ratios observed during ORC[3], which also indicates that higher mixing ratios were typically observed during campaign 2. The peak in the distribution of mixing ratios during campaign 1 was (4.2 ± 0.1) pptv, with a small difference between that of (4.3 ± 0.1) pptv during the daytime (~0600 to ~1900 hours) and that of (4.1 ± 0.1) pptv during the nighttime. During campaign 2, the peak in the distribution of mixing ratios was (5.2 ± 0.1) pptv, with a more pronounced difference between daytime and nighttime observations compared to campaign 1, with a value of (5.7 ± 0.1) pptv during the daytime and (4.6 ± 0.2) pptv during the nighttime. Diurnal profiles for glyoxal are shown in Figure 6. A weak diurnal profile is evident in the data, with the peak observed after midday and the mean daytime glyoxal mixing ratio slightly higher than the mean during the nighttime (4.3 pptv vs 4.0 pptv for campaign 1; 5.9 pptv vs 5.0 pptv for campaign 2).

A greater variability of glyoxal with wind direction was observed during campaign 2 compared to campaign 1, and a higher mean mixing ratio during campaign 2 associated with air masses arriving at the CVAO site from the east and greater variability of mixing ratios in air masses arriving from the north east (Figure S4). Figure S5 shows the probability distribution functions for the observed glyoxal mixing ratios separated by the air mass origin as indicated by ten-day back trajectory calculations performed using the NAME dispersion model (Ryall et al., 2001) using the technique described by Carpenter et al. (2010). Most of the data correspond to air masses originating from African coastal regions or a combination of European and African coastal regions. A Student's $t$-test indicates that there is no significant difference, at the $p < 0.05$ probability level, between data corresponding to air mass histories dominated by African coastal regions and data corresponding to the air masses dominated by a combination of European and African coastal regions. These results may imply some degree of continental influence on glyoxal abundance at CVAO. Given the short glyoxal lifetime, this influence could be via oxidation of precursors of continental origin in air masses influenced by African and/or European emission sources. A recent study examining sources and sinks of acetaldehyde in the remote Pacific atmosphere demonstrated the potential for a substantial secondary source of acetaldehyde in remote regions,





missing from models (Wang et al., 2019). The potential for a similar glyoxal source in the remote atmosphere under continental outflow warrants further investigation.

**4. Model simulations**

Model calculations were performed using the Dynamically Simple Model of Atmospheric Chemical Complexity (DSMACC), described in detail by Emmerson and Evans (2009) and Stone et al. (2010). DSMACC represents a zero-dimensional model framework and uses the Kinetic Pre-Processor (KPP) (Sandu and Sander, 2006) with the organic chemistry scheme described by the Master Chemical Mechanism (MCM, v3.2) (http://mcm.leeds.ac.uk/MCM/home.htt; Jenkin et al., 2003; Saunders et al., 2003) and IUPAC recommendations for kinetics of inorganic reactions (http://iupac.pole-ether.fr/; Atkinson et al., 2004; Atkinson et al., 2006). The MCM describes the near explicit oxidation schemes for 143 primary species, leading to ~6700 species interacting in ~17,000 reactions, and represents the most detailed and comprehensive chemistry scheme available for modelling tropospheric composition. The base model used in this work contains oxidation chemistry for methane, ethane, propane, *n*-butane, *iso*-butane, *n*-pentane, *iso*-pentane, *n*-hexane, ethene, propene, acetylene, isoprene, benzene, and toluene, incorporating ~1400 species in ~4300 reactions.

Photolysis rates in the model were calculated by the Tropospheric Ultraviolet and Visible (TUV) Radiation Model (http://cprm.acd.ucar.edu/Models/TUV/) and scaled to measurements of $NO_2$ photolysis frequencies ($jNO_2$), or to measurements of $jO(^1D)$ for $O_3$ photolysis, made by spectral radiometry using an Ocean Optics high resolution spectrometer (QE65000) coupled via fibre optic to a $2\pi$ quartz collection dome. The calculated photolysis frequencies for glyoxal use absorption cross-sections determined by Volkamer et al. (2005) and quantum yields recommended by the NASA Panel for Data Evaluation (http://jpldataeval.jpl.nasa.gov/; Sander et al., 2011).

Heterogeneous loss of $HO_2$ and glyoxal to aerosol surfaces was represented in the model by parameterisation of a first-order loss process to the aerosol surface (Schwarz, 1986):

$$k' = \left( \frac{r}{D_g} + \frac{4}{\gamma_x c_g} \right)^{-1} A \qquad \text{(Equation 5)}$$

where $k'$ is the first-order rate coefficient for heterogeneous loss, $r$ is the aerosol particle effective radius, $D_g$ is the gas phase diffusion coefficient (Equation 6), $\gamma_x$ is the uptake coefficient for species X, $c_g$ is the mean molecular speed (Equation 7), $A$ is the aerosol surface area per unit volume. $D_g$ is given by:

$$D_g = \frac{3}{8 N_A d_g^2 \rho_{air}} \sqrt{\frac{R T m_{air}}{2\pi} \left( \frac{m_g + m_{air}}{m_g} \right)} \qquad \text{(Equation 6)}$$

where $N_A$ is Avogadro's number, $d_g$ is the diameter of the gas molecule, $\rho_{air}$ is the density of air, $R$ is the gas constant, and $m_g$ and $m_{air}$ are the molar masses of gas and air, respectively. $c_g$ is given by:



$$c_g = \left( \frac{8RT}{\pi M_w} \right)^{1/2}$$

(Equation 7)

where $T$ is the temperature and $M_w$ is the molecular weight of the gas. The aerosol surface area in the model is constrained to previous measurements of dry aerosol surface area at the Cape Verde Observatory, corrected for differences in sampling height between the aerosol and glyoxal measurements and for aerosol growth under humid conditions (Allan et al., 2009; Muller et al., 2010; Whalley et al., 2010). For $HO_2$, $\gamma_{HO2}$ = 0.028 was used, based on the mean value reported by the parameterisation by Macintyre and Evans (2011), and consistent with values obtained in the laboratory for uptake of $HO_2$ onto aqueous inorganic salt aerosols (George et al., 2013). For glyoxal, $\gamma_{CHOCHO}$ = 0.001 was used (Liggio et al., 2005; Volkamer et al., 2007; Washenfelder et al., 2011; Li et al., 2014). Model sensitivity to aerosol uptake for glyoxal is discussed in Section 5.

An additional first-order loss process for each species in the model was also included to represent deposition processes, preventing the unrealistic build-up of model-generated intermediates in the box. Deposition velocities used in the model were 0.30 cm s$^{-1}$ for glyoxal (Volkamer et al., 2007; Huisman et al., 2011; Washenfelder et al., 2011; Li et al., 2014), 0.33 cm s$^{-1}$ for HCHO and other aldehydes (Brasseur et al., 1998), 1.00 cm s$^{-1}$ for $H_2O_2$ (Junkermann and Stockwell, 1999) and 0.90 cm s$^{-1}$ for organic peroxides (ROOH) (Junkermann and Stockwell, 1999), with a fixed boundary layer height of 713 m (Carpenter et al., 2010). For all other species, the deposition rate was set to be equivalent to a lifetime of approximately 24 h. Model sensitivity to this parameter has been discussed in previous work (Stone et al., 2010; Stone et al., 2014), with limited sensitivity displayed by modelled concentrations of radical species, and is discussed with reference to glyoxal in Section 5.

All measurements were merged onto a one-hour timebase, with model calculations for glyoxal performed if observations of pressure, temperature, water vapour concentration, $O_3$, CO, $NO_x$, VOCs and glyoxal were available for a specified timepoint, leading to a total of 872 independent simulations. For each time point, observed species are constrained to their measured value and kept constant throughout the model run. Concentrations of $CH_4$ and $H_2$ were kept constant at values of 1770 ppb (ftp://aftp.cmdl.noaa.gov/data/trace_gases/ch4/flask/surface/, Dlugokencky et al., 2014) and 550 ppb (Ehhalt and Rohrer, 2009; Novelli et al., 1999), respectively, with all other species for which observations were not available set initially to zero. A summary of species used to constrain the model is given in Table 1. Nitrogen oxides (NO, $NO_2$, $NO_3$, $N_2O_5$, HONO and $HO_2NO_2$) were constrained using the method described by Stone et al. (2010). Briefly, the model is initialised with the observed concentration of one nitrogen oxide species, in this case $NO_2$, and the concentrations of all nitrogen oxide species are allowed to vary according to their photochemistry as the model runs forwards. At the end of each 24 h period in the model, the calculated concentration of the nitrogen oxide species used to initialise the model is compared to its observed concentration, and the concentrations of all nitrogen oxide species fractionally increased or decreased such that the calculated concentration of the constrained species is equal to its observed concentration.





For each time point for which observations were available, the model is integrated forwards in time with diurnally
varying photolysis rates until a diurnal steady state is reached, typically requiring between 5 and 10 days. Once a
diurnal steady state has been reached, the concentrations of all species in the model and rates of all reactions are
output for the time of day corresponding to the time at which the observations used to constrain the model were
made.

## 5. Evaluation of Model Performance

### 5.1 Base model run

Figures 7 and 8 show the model performance for glyoxal during the two ORC[3] measurement campaigns, indicating a
general tendency to underpredict the glyoxal observations during both campaigns. The probability distribution
functions for the modelled to observed ratios for glyoxal, separated by the air mass origin as described above for
Figure S5, are shown in Figure S6. A Student's $t$-test indicates that there is no significant difference, at the $p < 0.05$
probability level, between the modelled to observed ratios corresponding to air mass histories dominated by African
coastal regions and data corresponding to the air masses dominated by a combination of European and African coastal
regions, as also shown for the observed concentrations in Section 3.

Figure 9 shows the average observed and modelled diurnal profiles for each measurement campaign. During the first
campaign, the average observed diurnal profile ranges from 3.8 pptv at night to 5.4 pptv during the day. For the second
campaign, daytime mixing ratios are typically higher, with the average observed diurnal profile ranging from a
minimum of 3.4 pptv at night to a maximum of 7.7 pptv during the day. For both campaigns, the model underpredicts
the observed glyoxal mixing ratios, with minima in the modelled glyoxal diurnal profiles of 0.5 pptv and 0.3 pptv for
the first and second measurement campaigns, respectively, occurring at approximately 0700 hours for both
campaigns, and maxima of 1.9 pptv and 1.7 pptv around midday for campaigns 1 and 2, respectively. The modelled
glyoxal mixing ratios, for both campaigns, display greater diurnal variation than is apparent in the observations, with
the model underprediction indicating either an overestimation of sinks or an underestimation of sources.

Diurnally averaged budget analyses for glyoxal are shown in Figure 10, with the budgets around midday (1100 to 1300
hours) shown in Figure 11. The general trends in the budgets are similar between the two campaigns and the following
discussion focuses on the average behaviour for the two campaigns combined. The dominant sources of glyoxal in the
model during the day are the reactions of OH with glycolaldehyde (2-hydroxyacetaldehyde, $HOCH_2CHO$) and acetylene
($C_2H_2$). Glycoaldehyde is produced primarily in the model following the oxidation of ethene ($C_2H_4$), and the reaction
between glycoaldehyde and OH represents, on average, 39 % of the total glyoxal production around midday, while
the reaction of acetylene with OH represents, on average, 35 % of the total glyoxal production around midday. There
is also significant production of glyoxal from the reaction of OH with the peroxide species $HCOCH_2OOH$, generated
following a minor channel (5 %) in the OH-initiated oxidation of acetaldehyde in reactions R1-R4, which represents 11
%, on average, of the total glyoxal production around midday.





$$CH_3CHO + OH \rightarrow HCOCH_2 + H_2O \quad (R1)$$

$$HCOCH_2 + O_2 + M \rightarrow HCOCH_2O_2 + M \quad (R2)$$

$$HCOCH_2O_2 + HO_2 \rightarrow HCOCH_2OOH + O_2 \quad (R3)$$

$$HCOCH_2OOH + OH \rightarrow CHOCHO + OH + H_2O \quad (R4)$$

6 While observations of $C_2H_2$ and $C_2H_4$ are available during the campaigns, and provide a constraint on the two dominant model pathways for glyoxal production, there is no such observational constraint available for the acetaldehyde

8 mixing ratio and its subsequent impact on glyoxal production. Results presented thus far include only acetaldehyde generated in the model following the oxidation of measured VOCs. For the first campaign, the mean model simulated

10 acetaldehyde mixing ratio was (14.2 ± 6.5) pptv (median 15.0 pptv), while that for the second campaign was (15.0 ± 5.6) pptv (median 16.3 pptv). Previous observations of acetaldehyde reported at the measurement site (Carpenter et

12 al., 2010; Read et al., 2012) have indicated mixing ratios on the order of several hundred pptv, while more recent airborne measurements made in the region of Cape Verde during the AToM campaign (Apel et al., 2019; Wang et al.,

14 2019) suggest lower mixing ratios of ~200-300 pptv in between the surface and 200 m altitude in August. The impact of acetaldehyde on the model is discussed further in Section 5.2.

Several reactions comprise the remaining source term for glyoxal, notably including those involving oxidation products

18 of benzene and toluene (for example, the MCM species BZOBIPEROH and TLOBIPEROH shown in Figures 10 and 11), with each providing a small contribution to the total. At night, the dominant sources of glyoxal are reactions of

20 intermediates generated by $O_3$- or $NO_3$-initiated oxidation of isoprene (> 80 % of the total) and $NO_3$-initiated oxidation of toluene.

Loss of glyoxal in the model is dominated by photolysis, which represents 50 % of the total sink term around midday

24 (note that the three possible photolysis channels in the model are shown individually in Figures 10 and 11), followed by reaction with OH (37 % of the midday sink) and uptake to aerosol (12 %). Nighttime losses are dominated by uptake

26 to aerosol and reaction with $NO_3$. The increase in the observed to modelled ratio for glyoxal at night may therefore result from uncertainties in the diurnal behaviour of the aerosol surface area, with the uptake of glyoxal to aerosols

28 potentially overestimated at night.

30 Modelled concentrations of OH and $HO_2$ are similar to observations and model calculations reported previously at the Cape Verde Atmospheric Observatory (Whalley et al., 2010; Vaughan et al., 2012), with diurnal maxima of ~$9 \times 10^6$ cm$^-$

32 $^3$ for OH and ~$5 \times 10^8$ cm$^{-3}$ for $HO_2$. Since OH is involved in both the production and loss of glyoxal, the modelled glyoxal concentrations are relatively insensitive to the concentrations of OH and $HO_2$ in the model. Thus, while previous

34 measurement campaigns in Cape Verde (Read et al., 2008; Whalley et al., 2010; Stone et al., 2018) have demonstrated the importance of halogen chemistry for understanding observations of $O_3$, OH and $HO_2$, the model output for glyoxal

36 is not sensitive to halogen chemistry, and model simulations presented here do not include halogens.





### 5.2 Impact of acetaldehyde

2    The OH-initiated oxidation of acetaldehyde leads to the production of glyoxal following H-atom abstraction from the

methyl group of acetaldehyde (R1-R4), which is a minor channel in the reaction between OH and acetaldehyde with a

4    recommended branching ratio of 0.05 (Atkinson et al., 2004). The dominant reaction channel in OH + $CH_3CHO$, with a

branching ratio of 0.95, leads to the production of acetylperoxy radicals ($CH_3C(O)O_2$) and does not result in production

6    of glyoxal. However, the minor channel has a potentially significant impact on glyoxal and for the base model run,

unconstrained to acetaldehyde and thus using acetaldehyde concentrations generated in the model through oxidation

8    of other measured VOCs, the minor channel in OH + $CH_3CHO$ contributes 11 % of the total glyoxal production around

midday (Section 5.1). The production of glyoxal following the OH-initiated oxidation of acetaldehyde is thus sensitive

10    to both concentration of acetaldehyde and the branching ratio for OH + $CH_3CHO$.

12    The branching ratio for OH + $CH_3CHO$ adopted in the MCM uses the current IUPAC recommendation (Atkinson et al.,

2004), which is based on experimental studies at room temperature by Cameron et al. (2002) and Butkovskaya et al.

14    (2004). Cameron et al. used pulsed laser photolysis with direct detection of $CH_3CO$, $CH_3$ and H-atom products at a total

pressure of 60 Torr to determine the branching ratios for channels producing $CH_3CO + H_2O$, $CH_2CHO + H_2O$, $CH_3$ +

16    $HC(O)OH$ and $H + CH_3C(O)OH$, and concluded that neither H nor $CH_3$ are produced directly in the reaction between OH

and acetaldehyde at detectable concentrations, with production of $CH_3CO$ (resulting from H-atom abstraction from

18    the aldehyde group) at a yield of (0.93 ± 0.18) and an upper limit of 0.25 for production of $CH_2CHO$ (resulting from H-

atom abstraction from the methyl group). Butkovskaya et al. used a high pressure (200 Torr) turbulent flow reactor

20    with chemical ionisation mass spectrometry, which enabled direct detection of the $CH_2CHO$ radical, giving a branching

ratio for $CH_2CHO$ production of $(0.051^{+0.024}_{-0.017})$ with a total $H_2O$ yield of (0.977 ± 0.0447), and evidence for production

22    of glyoxal from $CH_2CHO$ in the presence of $O_2$. The relative uncertainty in the branching ratio for H-atom abstraction

from the methyl group of acetaldehyde is thus significant, and has yet to be investigated over a range of temperatures

24    and pressures.

26    The base model run (Section 5.1) was unconstrained to acetaldehyde and contains only the acetaldehyde generated

in the model following the oxidation of measured VOCs, with typical acetaldehyde mixing ratios of ~10-20 pptv and

28    mean values of (14.2 ± 6.5) pptv and (15.0 ± 5.6) pptv for the first and second campaigns, respectively. For the base

model run, over 90 % of the modelled acetaldehyde was generated following the oxidation of ethane and propene,

30    and the model neglects potential production of acetaldehyde from oxidation of higher alkanes, as well as ocean

sources of acetaldehyde, which have been shown to be significant (Singh et al., 2003; Lewis et al., 2005; Heald et al.,

32    2008; Millet et al., 2010; Carpenter et al., 2012; Read et al., 2012; Yang et al., 2014). Measurements of higher > C5

alkanes are not available, and are found to be frequently below detection limits at the CVAO site. Recent airborne

34    measurements made near Cape Verde during the ATom campaign have indicated acetaldehyde mixing ratios of ~200–

300 pptv during August 2016, measured between the ocean surface and ~200 m altitude (Apel et al., 2019; Wang et

36    al., 2019). The impact of acetaldehyde on glyoxal is thus likely to be underestimated by the base model run. Here, we

investigate the model sensitivity to the mixing ratio of acetaldehyde using the recommended branching ratio of 0.05

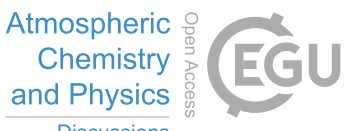

(Atkinson et al., 2004) for CH₃CHO + OH → CH₂CHO + H₂O (R1), but note that the modelled glyoxal is also sensitive to the branching ratio, and that, in terms of glyoxal production, a reduction of the branching ratio to 0.04 is equivalent to a 20 % reduction in the acetaldehyde mixing ratio.

Figure 12 shows the sensitivity of glyoxal to the acetaldehyde mixing ratio in the model and compares the model output for glyoxal for the base model run (unconstrained to acetaldehyde) to a series of model runs in which acetaldehyde was constrained to a fixed mixing ratio between 25 and 500 pptv throughout each model run. There is a substantial impact on the modelled glyoxal during the day, with the modelled glyoxal approaching the observed mixing ratios during the day in the first campaign on constraint to 200 pptv acetaldehyde which is consistent with the acetaldehyde observations made during the ATom campaign (Apel et al., 2019; Wang et al., 2019). For the model constrained to 200 pptv acetaldehyde, the production of glyoxal following OH + CH₃CHO represents 56 % of the total glyoxal production at midday, with this mechanism dominating the glyoxal production for acetaldehyde mixing ratios above 100 pptv. While significantly higher mixing ratios of acetaldehyde are required to reproduce the observed glyoxal at night and during the day in the second campaign, the model results imply that the acetaldehyde mixing ratio and branching ratio for OH + CH₃CHO are critical to accurately simulate glyoxal in the tropical marine environment.

### 5.3 Impact of terpenes

Measurements at the Cape Verde Atmospheric Observatory indicate a total of approximately 1 pptv of monoterpenes, comprising α-pinene, β-pinene, limonene, myrcene, Δ-3-carene and ocimene (Hackenberg, 2015). Total monoterpene average mixing ratios of between 0.05 and 5 pptv have been observed in the Atlantic marine boundary layer (Hackenberg et al., 2017), and larger average mixing ratios of 17 pptv in the North Atlantic (Kim et al., 2017). Oxidation of monoterpenes has the potential to reconcile the discrepancies between the observed and modelled glyoxal and warrants further investigation.

### 5.3.1 Local terpene sources

The observed monoterpene mixing ratios likely result from local sources owing to the high reactivity of monoterpenes and their rapid photochemical removal from the atmosphere. The MCM contains full oxidation schemes for the monoterpenes α-pinene, β-pinene and limonene (Saunders et al., 2003), and for the sesquiterpene β-caryophyllene (Jenkin et al., 2012). Impacts of these compounds on the modelled glyoxal mixing ratios were investigated by constraining the model to a range of fixed concentrations of each terpene, for model runs unconstrained to acetaldehyde.

The model sensitivity to the monoterpene species α-pinene and β-pinene is shown in Figure S7. While there are some increases in the modelled glyoxal mixing ratios during the day, there are limited impacts at night, and even during the day mixing ratios of α-pinene in excess of 10 pptv (i.e. significantly in excess of observed abundance (Hackenberg, 2015)) are required to reproduce the glyoxal observations. Yields of glyoxal from β-pinene, and limonene (not shown),



are lower than those from α-pinene, with almost no change in the modelled glyoxal mixing ratio on inclusion of 10 pptv limonene in the model.

Glyoxal yields from the sesquiterpene β-caryophyllene, however, are higher than those for the monoterpenes. Figure S8 shows the model sensitivity to β-caryophyllene, indicating that low mixing ratios of β-caryophyllene in the model can significantly enhance the modelled glyoxal during the day. The inclusion of ~1 pptv and ~2.5 pptv β-caryophyllene for the first and second campaigns, respectively, can reproduce the daytime observations of glyoxal. There is also significant production of glyoxal at night resulting from β-caryophyllene chemistry, in contrast to the monoterpenes, although higher β-caryophyllene mixing ratios (~ 5 pptv in the early evening and ~ 10 pptv in the early hours of the morning) are required to reproduce the nighttime glyoxal observations. While such variation in the mixing ratio of a species with temperature dependent emissions from a biogenic source is not inconceivable, particularly if the principal removal mechanism is reaction with OH, the potential role for marine sources of sesquiterpenes in the chemistry at Cape Verde is uncertain.

Thus, while monoterpenes have the potential to produce glyoxal, the impacts are limited, and monoterpene mixing ratios higher than those observed in the Atlantic Ocean are required to explain the discrepancies between the modelled and observed glyoxal at night. In addition, given the short lifetimes of monoterpene species, significant local emissions would be required to sustain such high mixing ratios. Lower mixing ratios of the sesquiterpene β-caryophyllene are able to reproduce the glyoxal observations at night, but local sources are unknown. There is only limited indirect evidence that may support the presence of sesquiterpenes in the marine boundary layer, including measurements of markers for β-caryophyllene products in organic aerosol samples from the summertime Arctic marine atmosphere (Fu et al., 2013). However, to our knowledge there have been no direct observations of sesquiterpenes in the marine boundary layer atmosphere.

### 5.3.2 Remote monoterpene sources

Production of glyoxal from monoterpenes and the sesquiterpene β-caryophyllene involves a number of intermediates in multiple reactions (production following OH + α-pinene requires at least 19 reactions). Phytoplankton blooms in the South Atlantic Ocean have been shown to result in high monoterpene emissions and can lead to atmospheric mixing ratios of ~200 pptv (Yassaa et al., 2008). Similarly, observations from the North Atlantic marine boundary layer have shown monoterpene mixing ratios as large as 100 pptv (Kim et al., 2017). There is thus potential for production of high concentrations of glyoxal in air masses transported from regions with high monoterpene emissions, which contain significant concentrations of terpene oxidation intermediates but, by the time the air mass has been transported away from the emission source, low concentrations of the parent monoterpene.

In order to investigate the potential for such a scenario to explain enhanced glyoxal abundances at CVAO, the model was constrained to the average mixing ratios of species observed between 1000 and 1400 hours at the Cape Verde



measurement site (Table 2) and initialised with, but not constrained to, a high monoterpene mixing ratio (100 pptv α-pinene and 50 pptv β-pinene) characteristic of a region associated with a phytoplankton bloom (Yassaa et al., 2008). The model was initialised at midday and then run forwards in time for several days to represent the chemical evolution of an air mass passing over an area with high monoterpene emissions, such as a phytoplankton bloom.

| Parameter | Model constraint |
|---|---|
| Pressure / hPa | 1013 |
| Temperature / K | 298 |
| Water vapour / ppm | 26594 |
| $O_3$ / ppb | 25.7 |
| $NO_2$ / pptv | 22.1 |
| NO / pptv | 5.9 |
| CO / pptv | 79.0 |
| $C_2H_6$ / pptv | 467.0 |
| $C_3H_8$ / pptv | 18.5 |
| $IC_4H_{10}$ / pptv | 1.5 |
| $NC_4H_{10}$ / pptv | 2.7 |
| $C_5H_8$ / pptv | 0.5 |
| $IC_5H_{12}$ / pptv | 0.8 |
| $NC_5H_{12}$ / pptv | 0.5 |
| $NC_6H_{14}$ / pptv | 8.9 |
| $C_2H_4$ / pptv | 32.1 |
| Benzene / pptv | 6.1 |
| Toluene / pptv | 0.7 |
| $C_2H_2$ / pptv | 35.9 |

Table 2: Conditions used to constrain the model run to demonstrate the downwind impact on glyoxal of remote monoterpene sources. Mixing ratios are the midday mean values of all data during both ORC[3] measurement campaigns.

The impact of an air mass passing over a region with high monoterpene emissions on the monoterpene and glyoxal mixing ratios for initial monoterpene mixing ratios of 100 pptv α-pinene and 50 pptv β-pinene (Yassaa et al., 2008) is shown in Figure S9. The monoterpenes are rapidly consumed (in less than four hours), but the glyoxal mixing ratios are significantly elevated compared to equivalent model runs with no monoterpene input owing to the chemistry of the oxidation products of the parent monoterpenes. The temporal profiles for glyoxal exhibit diurnal behaviour, maximising each day, with overall maxima observed ~30 hours after initialisation of the model, with subsequent days displaying lower maxima compared to the previous day.



2    The monoterpene oxidation products involved in the generation of glyoxal consist of a number of peroxide and aldehyde species, the modelled concentrations of which, in the absence of a constant monoterpene source, are

4    influenced by the deposition rates applied to the model-generated oxidation intermediates. Figure S9 shows the impact of the deposition rate applied in the model to the evolution of glyoxal. The modelled mixing ratios of glyoxal

6    are strongly influenced by the deposition rates applied in the model, with slower deposition rates leading to longer lifetimes of the oxidation intermediates and thus higher mixing ratios of glyoxal. In the absence of any initial

8    monoterpene input, the modelled glyoxal mixing ratio is ~2.2 – 2.4 pptv after ~30 hours, and displays little sensitivity to the deposition rates in the model. For the model initialised with 100 pptv of α-pinene and 50 pptv of β-pinene, the

10   modelled glyoxal varies between 2.7 pptv (using the standard deposition rates used in the model) and 3.6 pptv (with deposition rates decreased by a factor of four).

Thus, while air masses originating from areas with high monoterpene emissions have the potential to produce elevated

14   glyoxal mixing ratios downwind of the region with high monoterpene concentrations owing to the action of the monoterpene oxidation products, the impacts are highly dependent on the deposition lifetimes applied to the model-

16   generated intermediates. The deposition rates of such species are highly uncertain, but can have significant impact on the modelled glyoxal.

**5.4 Impact of physical processes**

20   The model contains a first-order loss process for all unconstrained species to represent deposition and physical losses to prevent the build-up of unrealistic concentrations of model-generated intermediates, which, for most species, is

22   set to give a lifetime of ~24 h. Given the significance of model-generated intermediates for the production of glyoxal, there is the potential for modelled glyoxal concentrations to be affected by the deposition rates implemented in the

24   model.  Figure S10 shows the model sensitivity to the deposition rates, and uptake of glyoxal to aerosol.

26   Increased deposition rates lead to a reduction in the modelled glyoxal mixing ratio, as the concentrations of intermediates such as glycoaldehdye and the peroxide $HCOCH_2OOH$ produced following the oxidation of acetaldehyde

28   are impacted by the change in deposition rates. However, the deposition rates applied in the model would need to be reduced significantly (by more than a factor of four) to reconcile the modelled glyoxal with the observations. A

30   reduction in the aerosol uptake of glyoxal does improve the agreement between the modelled and observed glyoxal, particularly at night and in the early morning, potentially indicating that the constant aerosol surface area

32   implemented in the model is uncertain and an over-simplistic approximation.  However, while there is uncertainty associated with the aerosol surface area implemented in the model, the sensitivity of the modelled glyoxal to changes

34   in the rate of aerosol uptake is not sufficient to reconcile the model with the observations.



## 6. Conclusions and implications

We have made the first in situ measurements of glyoxal in the tropical remote marine boundary layer using a sensitive laser-induced phosphorescence (LIP) technique, with a temporal resolution of a few minutes. Previous remote marine glyoxal observations have mainly been based on remotely-sensed measurements or coarse temporal averages using off-line methods. LIP glyoxal measurements were made continuously over two 4-week campaigns during June-July and August-September 2014 at the Cape Verde Atmospheric Observatory in the tropical North Atlantic. The sensitive LIP technique achieved a limit of detection of ~1 pptv, allowing measurement of 24-hour average glyoxal mixing ratios of 4.9 pptv and 6.3 pptv during the first and second campaigns respectively. The overall maximum observed mixing ratio was 36.3 pptv, measured on 22nd June 2014 during campaign 2. A weak diel variation in measured glyoxal was observed, particularly during campaign 1 (daytime median 4.3 ± 0.1 pptv; nighttime median 4.1 ± 0.2 pptv), with slightly larger diurnal variability during campaign 2 (daytime median 5.7 ± 0.1 pptv; nighttime median 4.6 ± 0.2 pptv). Our average glyoxal mixing ratios are consistent with offline HPLC measurements made at Cape Grim on the western Tasmania coast Lawson et al., 2015), and our highest observed mixing ratios during August-September are similar in magnitude to HPLC measurements in the South Pacific (23 ±8 pptv) from the same study. Our highest observed mixing ratios are also consistent with the lowest MAX-DOAS remote sensed mixing ratios in the northern hemisphere tropical East Pacific (Coburn et al., 2014). However, other previously measured MAX-DOAS glyoxal mixing ratios in the tropical East Pacific (range 40-140 pptv) (Sinreich et al., 2010) are substantially larger than our observed  mixing ratios in the tropical Atlantic.

We found no strong relationship between the observed glyoxal abundance and air mass origin. Back trajectory calculations show some evidence for greater variability in glyoxal abundances among air masses that have been influenced by coastal and continental Africa. During August and September greater variability in observed glyoxal with wind direction was observed, with higher median concentrations of 6.0 pptv associated with winds from the east, and lower concentrations of 4.9 pptv and 4.5 pptv from the north and northwest, respectively. During June, median mixing ratios associated with different wind sectors differed by less than 1 pptv.

The DSMACC box model with the explicit Master Chemical Mechanism, constrained by observed trace gas concentrations, significantly underpredicts the observed glyoxal concentrations during the day and night for measurements made during Jun-July and Aug-Sept. An investigation of the glyoxal budget in the model around local midday shows that the daytime glyoxal source is dominated by production from reactions of OH with glycoaldehyde and acetylene. During the day, a minor channel in acetaldehyde oxidation, initiated by H abstraction from the acetaldehyde methyl group, also contributes to glyoxal production. Increasing the acetaldehyde concentration in the model, which is likely underestimated in the base model run, significantly increases modelled glyoxal concentrations, with the mean observed-to-modelled ratios around midday (1100-1300 hours) improving from 3.2 and 4.2 for campaigns 1 and 2, respectively, for the base model run to 1.3 and 1.8 on constraint to 200 pptv acetaldehyde. This acetaldehyde mixing ratio is consistent with summertime near-surface concentrations measured from the AToM aircraft campaign in the region of Cape Verde.



2  The model underestimation of the observed glyoxal at night (nighttime (2200-0500 hours) mean observed-to-modelled ratio of 9.7) suggests there is also a missing non-photochemical glyoxal source in the model or potentially

4  an overestimate in model sink processes. Despite the importance of direct sea-air transfer for remote marine budgets of other oVOCs, such as acetone, acetaldehyde, and methanol, a similar source for glyoxal would seem unlikely due to

6  its high solubility and potential rapid consumption in surface ocean waters (Volkamer et al., 2009; Ervens and Volkamer 2010). However, eddy covariance measurements in the southern hemisphere tropical Pacific have shown net positive

8  sea-to-air glyoxal fluxes during nighttime, which are hypothesised to result from ozone-driven glyoxal production in the SML that out-competes the surface deposition flux (Coburn et al., 2014). Such a source cannot be ruled out during

10  the ORC[3] campaigns, and may explain at least some of the model underestimate. Ozone-driven reactions to produce glyoxal from idealised organic films in seawater have been shown from laboratory experiments (Zhou et al., 2014).

12  Further investigation using a coupled sea-air transfer model to investigate the potential impact of such a source is warranted.

We investigated the possible role for monoterpenes as glyoxal precursors using the DSMACC model. Observed

16  concentrations of monoterpenes at CVAO are insufficient to act as an appreciable source of glyoxal, and monoterpene concentrations larger than those observed during the ORC[3] campaign are required to explain the discrepancies

18  between the modelled and observed glyoxal. The model suggests that low mixing ratios (1-2.5 pptv) of the sesquiterpene β-caryophyllene would produce sufficient glyoxal to match observed concentrations, but there is no

20  direct evidence for the presence of sesquiterpenes at CVAO. Using the model to simulate the potential influence of a large monoterpene source, upwind of CVAO, showed the potential to produce elevated glyoxal mixing ratios

22  downwind of such a source via oxidation of monoterpene oxidation products. However, the magnitude of this potential source is highly sensitive to the deposition rates of the intermediate products, which are largely unknown.

Overall, our model results imply that our understanding of glyoxal in the remote marine boundary layer may be largely

26  controlled by our knowledge of the remote marine acetaldehyde budget and acetaldehyde oxidation chemistry, in particular the mixing ratio for acetaldehyde and the branching ratios for H-atom abstraction at different sites in

28  acetaldehyde, which are both associated with significant uncertainties. Given considerable uncertainties in our understanding of remote acetaldehyde sources, and a widespread underestimate of acetaldehyde in the remote

30  troposphere in models (Wang et al., 2019), these factors may limit our ability to constrain the glyoxal budget in the large-scale remote atmosphere at present. In addition, improved understanding of monoterpene sources in the

32  marine boundary layer and the influence of upwind monoterpene emissions in air masses sampled at CVAO may help constrain other potential influences on the glyoxal budget in this region. Further analysis using 3D chemical transport

34  modelling is needed to investigate further the potential for large-scale secondary production of glyoxal from VOC precursors in the remote marine atmosphere.





**Acknowledgements**

The authors would like to thank the Natural Environment Research Council (NERC) for funding the ORC$^3$ project (NE/K006665/1 and NE/K004980/1). DS would also like to thank NERC for the award of an Independent Research Fellowship (IRF) (NE/L010798/1). We would also like to thank NERC for the provision of a research studentship for DC. The authors acknowledge Luis Mendes, Instituto Nacional de Meteorologia e Geofísica, São Vicente (INMG), Mindelo, Cabo Verde, for his technical support of the measurements.



2 **Figures**

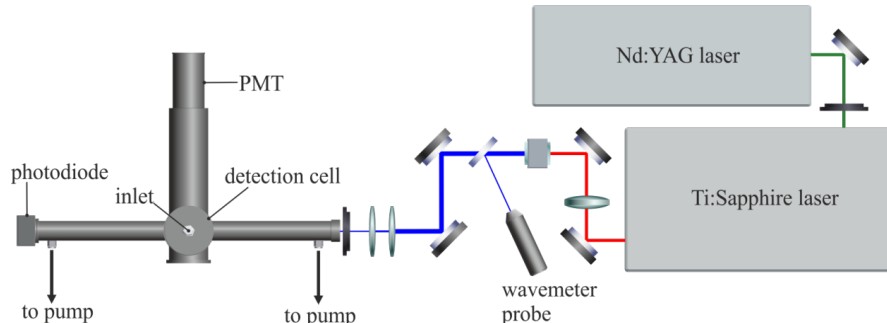

Figure 1: Plan view schematic of the glyoxal instrument, showing the arrangement of lasers, optics, and detection cell.





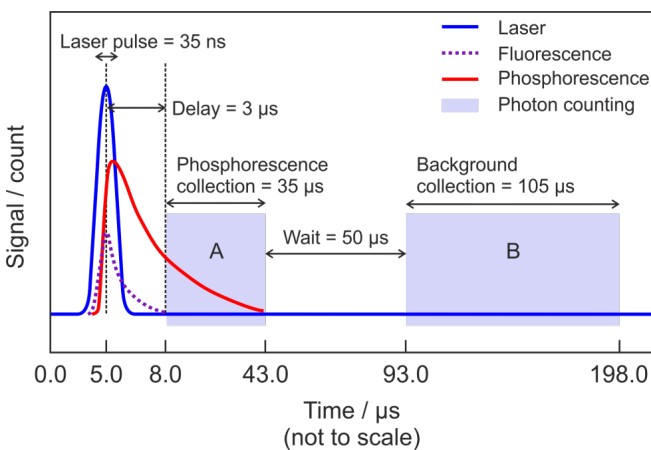

4 Figure 2: Schematic diagram showing the timing scheme for signal acquisition. The start of the cycle (time = 0.0 µs) is

defined by the laser trigger. Blue = Ti:Sapphire laser pulse at λ ~ 440 nm; dashed purple = short-lived fluorescence

6 from the cell anodising dye, optics, and interfering species (e.g. $NO_2$); red = glyoxal phosphorescence; light (not to

scale) blue shading = photon counting. Region A indicates the gate for measurement of the combined

8 phosphorescence signal, remaining laser scattered light signal and background signal, while region B indicates and the

gate for measurement of the background signal alone.





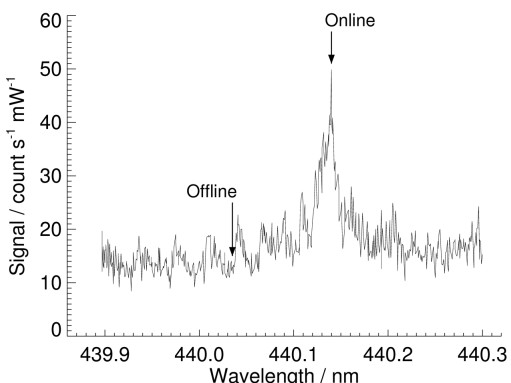

Figure 3: Laser-induced phosphorescence $\left(8_0^1\right)$ spectrum of ~ $8 \times 10^9$ molecule $cm^{-3}$ glyoxal at a cell pressure of 100

6    Torr, showing the online and offline wavelengths used for ambient glyoxal measurements. All background signals have

been removed from the data, with the signal thus representing glyoxal laser-induced phosphorescence for all

8    wavelengths shown. Each data point was integrated over 1 second (5000 laser pulses). The wavelength increment

between adjacent data points is 0.0008 nm.





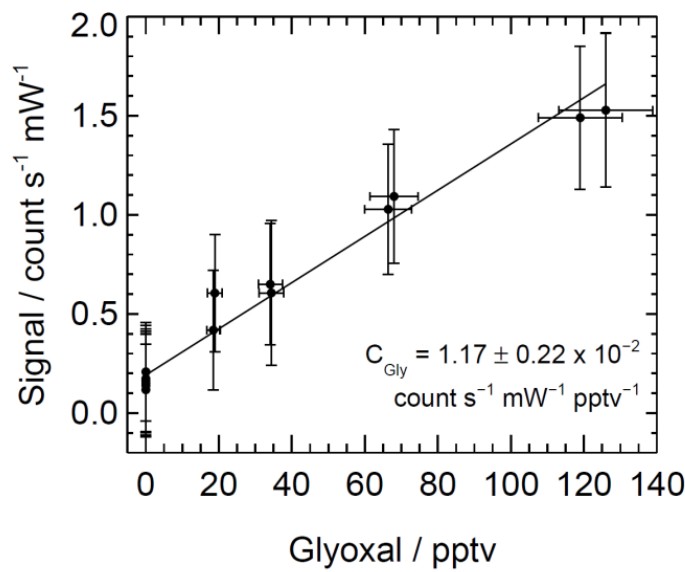

Figure 4: Typical calibration plot for glyoxal obtained during ORC[3], giving $C_{Gly}$ = (1.17 ± 0.22) × 10$^{-2}$ count s$^{-1}$ mW$^{-1}$ pptv$^{-}$

4      [1].





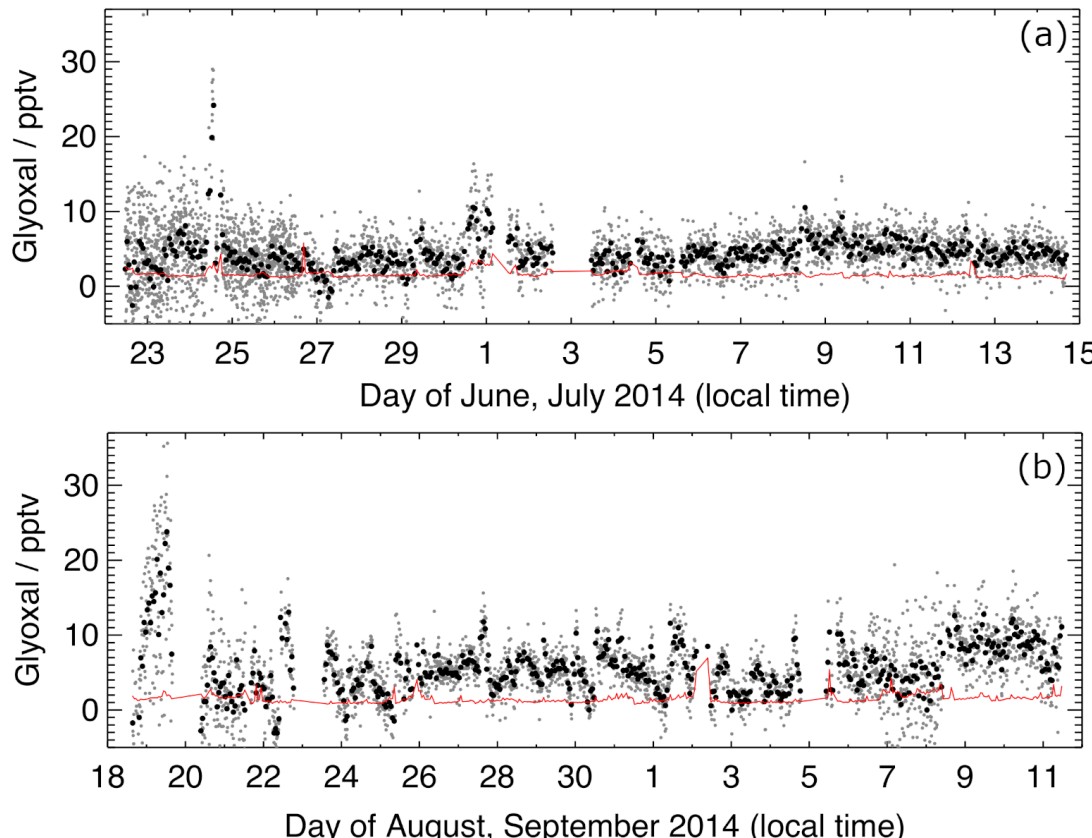

4 Figure 5: Time series of glyoxal data during (a) campaign 1 and (b) campaign 2. Black data points show 1-hour mean;

grey data points show the mean for a single measurement cycle (approximately 7 minute); red line shows limit of

6 detection.



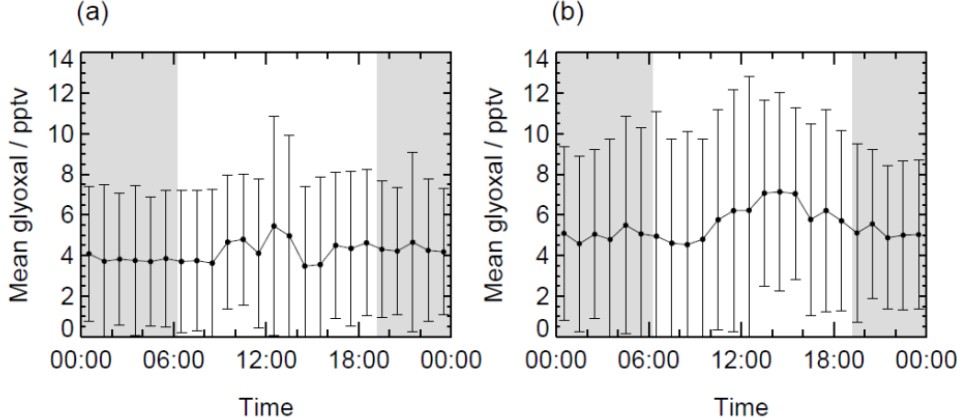

Figure 6: 1 hour average diurnal profiles of glyoxal during (a) campaign 1 and (b) campaign 2. Error bars represent the

4    1σ standard deviation of the measurements. Grey shading indicates hours of darkness.



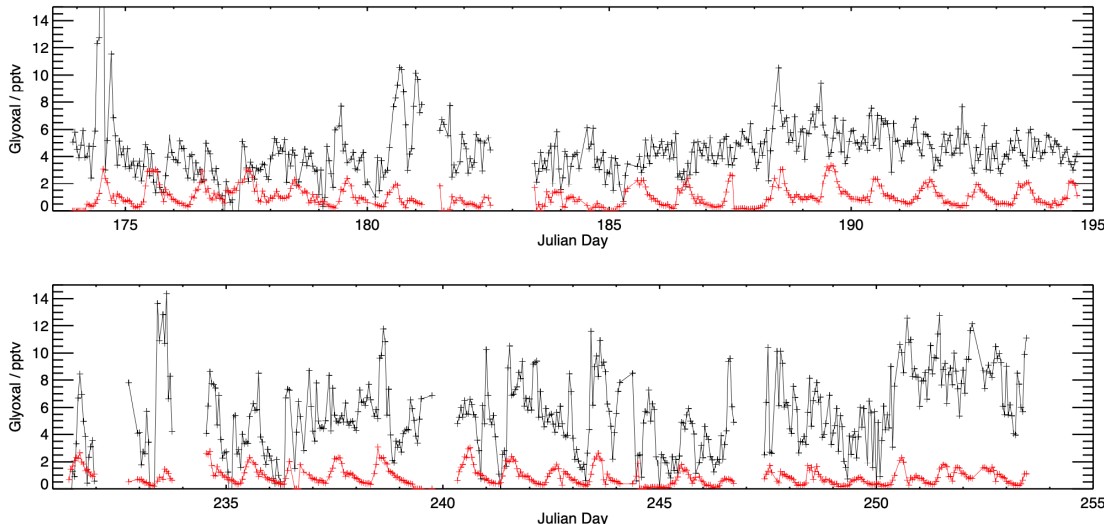

Figure 7: Time series of observed (black) and modelled (red) glyoxal during ORC[3]. Data are shown for model output

4    every 60 min. The upper panel shows data from the first measurement campaign (Julian days 173 to 195), with the

lower panel showing data from the second measurement campaign (Julian days 230 to 255).

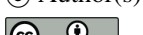



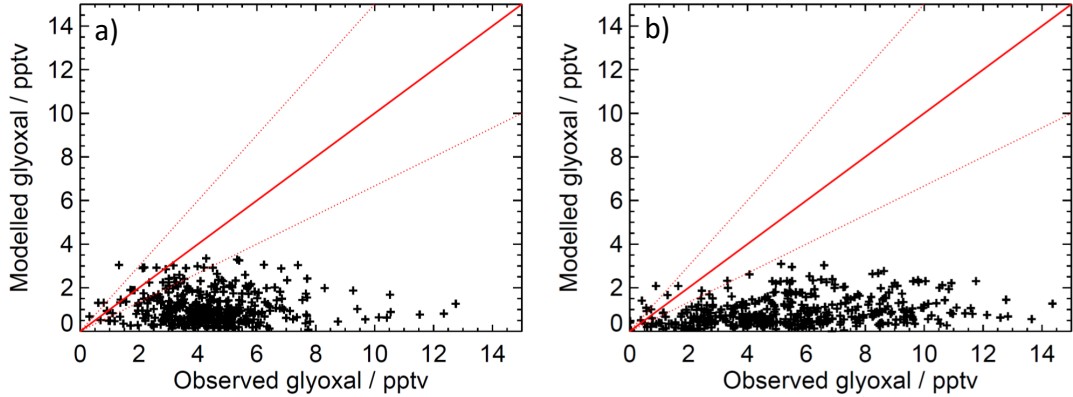

Figure 8: Modelled vs observed glyoxal for (a) the first measurement period and (b) the second measurement period

4     for the base model run.  Red lines show the 1:1 line with ± 50 % given by the dashed red lines.





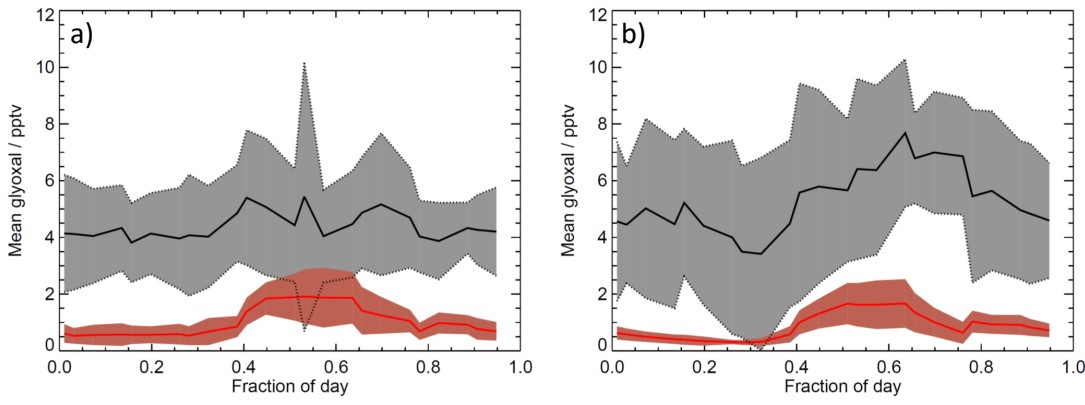

Figure 9: Average observed (black) and modelled (red) glyoxal diurnal profiles for a) the first measurement period and
4  b) the second measurement period for the base model run. Shaded regions show the 1σ variability of the data. Note
that the diurnal means include only data for which supporting measurements required to run the model are available
6  and thus show small differences from the diurnal means displayed in Figure 8.

2    Figure 10: Diurnally averaged glyoxal budgets for the base model run for both measurement periods combined.
Chemical names are as given by the MCM.





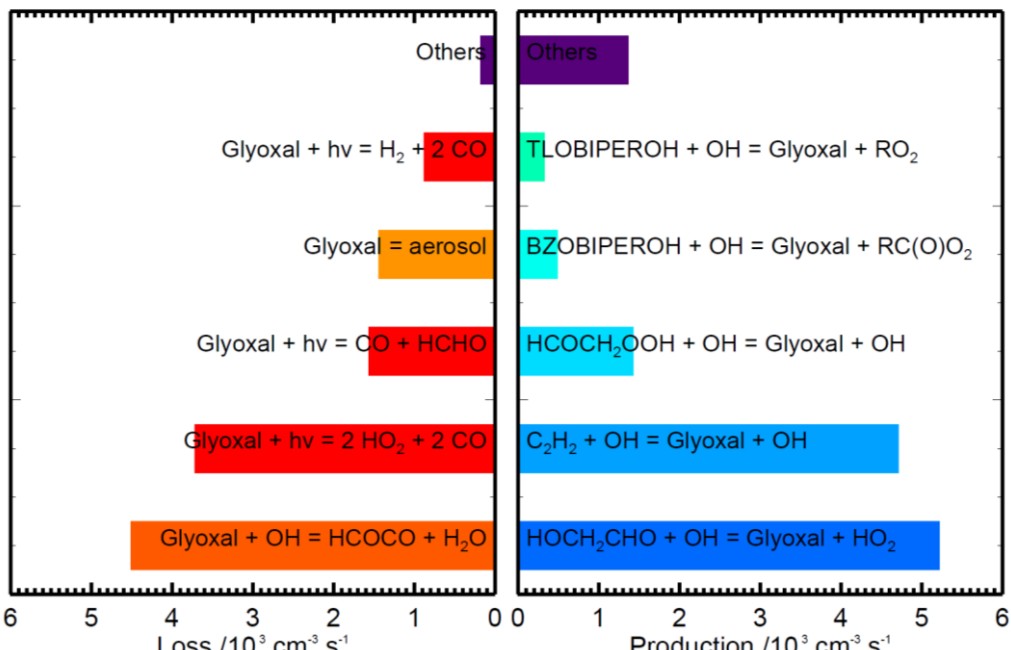

Figure 11: Midday (1100-1300 hours) average glyoxal budgets for the base model run for both measurement periods combined. Chemical names are as given by the MCM. Note that there are three photolysis channels contributing to the loss of glyoxal which are shown separately but constitute the dominant loss process overall.



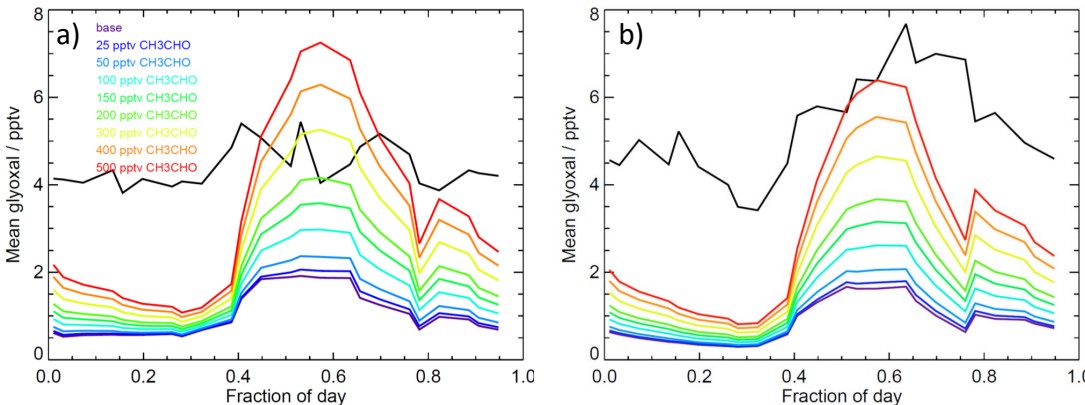

Figure 12: Average observed (black) and modelled glyoxal diurnal profiles showing the sensitivity of the model to

4   mixing ratios of acetaldehyde. The base model run includes only acetaldehyde produced within the model by the

oxidation of VOCs constrained in the model (Table 1) and has mean acetaldehyde mixing ratios of (14.2 ± 6.5) pptv

6   during campaign 1 (median 15.0 pptv) and (15.0 ± 5.6) pptv during campaign 2 (median 16.3 pptv). For model runs

performed to investigate the sensitivity of the model to acetaldehyde, the acetaldehyde mixing ratio in each

8   simulation was fixed to a constant mixing ratio between 25 and 500 pptv as indicated in the legend.



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
