# Peer review of "Observations and modelling of glyoxal in the tropical Atlantic marine boundary layer"

_Atmospheric Chemistry and Physics, 2021_

## Author Comment (AC1)

**Observations and modelling of glyoxal in the tropical Atlantic marine boundary layer – Response to Anonymous Referee #1**

We thank the reviewer for their positive and supportive comments on our manuscript. Below we respond to each specific comment in turn. Reviewer comments are shown in italic text, and our responses in normal text. Where we have added new text to the manuscript, this is summarised in bold text. Line numbers refer to the previously submitted original manuscript.

*P 7 line 12: what is trace heating tape?*
Trace heating tape is the name given to self-regulating heating cables which enable controllable heat to be applied to the sampling line in which the gases flow. The wording has been changed to:

**"The sample line is heated using self-regulating trace heating tape to enable controllable heat to be applied to the sample line."**

*P 7 line 28 and Figure 1. It would help the casual reader to put the pulse delay generator in the figure and to better explain the trigger and the pulse.*

Figure 1 is intended as a simple schematic that is not meant to be comprehensive to include all parts of the instrument. Rather than adding a box indicating the delay generator we have added some words to the Figure 1 caption as follows:

**"Not shown is the delay generator, which controls the timing of the experiment, namely the firing of the laser pulse and also the collection windows of the photon counter which measures both the phosphorescence signal and the background signal."**

The pulse sequence of the triggers from the pulse delay generator are already described in some detail between lines 26 and 36 and illustrated in Figure 2, which shows the timing sequence in detail. However, we have modified the sentence starting on line 26 to the following to provide some further clarity:

**"Figure 2 shows the timing scheme for signal acquisition. A delay generator (Berkeley Nucleonics Corporation, Model 555 Delay-Pulse Generator) is used to control the relative triggering times of several pieces of equipment, namely when the excitation laser fires and the two collection window of the photon counter which measures both the phosphoresence signal and the non-laser background signal."**

*Table 1: Data all look reasonable except n-hexane is surprisingly high. My guess is that there may be an issue with the measurement although it this will not impact any conclusions in the paper so it is minor.*

We thank for the reviewer for raising the potential issue with the n-hexane data. Upon checking it seems that there may have been an interference issue with the hexane data specifically (but not with other VOCs) associated with the GC particulate filter. Despite this problem, we note that the model results display almost no sensitivity to the assumed hexane concentration. We have carried out tests with the model and removing n-hexane from the simulations results in an average change to the simulated glyoxal of 0.018%. Due to this very small sensitivity and the problems with the data, we have elected to remove the n-hexane data from Table 2 and Figure S1 to avoid presenting potentially erroneous data, while the model results are unaffected.

*I'm not sure that using the MCM names for chemical species throughout the document is the best approach but leave it to authors' discretion.*

We appreciate this comment, but we feel that retaining use of the MCM species names is preferable. For simpler species the molecular formulae and names are the same in any case. For the aromatic oxidation products shown in Figures 10 and 11 and stated on Page 16, it is not clear that the overly long and complex IUPAC names would be helpful to the reader. The MCM names are traceable and at least retain some information on the parent identify of these species. We have added the following to the captions of Figures 10 and 11 to point the reader to further information on these species in the MCM mechanism:

**"For information on molecular formulae and details of production mechanisms of BZOBIPEROH and TLOBIPEROH, please see: http://mcm.york.ac.uk/roots.htt#aromatics."**

*P 12 line 2: Put dates here for the two respective campaigns*

Dates have been added:

**"The instrument was operational for 24 days and 22 nights during the first campaign (22nd June to 15th July 2014), and for 25 days and 21 nights during the second campaign (18th August to 15th September 2014)."**

*P 12 line 11. I would change sentence for clarification to "The maximum glyoxal mixing ratio of 36.3 pptv was observed during the first campaign on 22 June 2014; however, the 24 hour…."*

The sentence has been adjusted as suggested.

*Page 13 line 18 – update to: https://www2.acom.ucar.edu/modeling/tropospheric-ultraviolet-and-visible-tuv-radiation-model*

This has been updated.

*P 15 paragraph beginning with line 17: Maybe I missed it but do you discuss/speculate why the mixing ratios are higher during the second campaign? If not, this would be a good addition or if you don't know of any reasons why then perhaps state that.*

This is a good point, and we agree that hypothesising reasons for this difference would be useful to include. During August and September (Campaign 2) Cape Verde is subject to increased influence of North-easterly and Easterly air mass origins, with less influence from Westerly and North-westerly remote Atlantic air masses (see climatology in Carpenter et al., (2010)). This shift in influence appears to result in more variable and enhanced glyoxal concentrations in easterly and north-easterly air masses (Fig. S5), which have had recent influence from the African coast and continent.

We have added the following text to the manuscript (Page 15, Line 15):

**"Larger average glyoxal mixing ratios observed during Campaign 2 compared with Campaign 1 may be linked to a shift in air mass transport patterns to Cape Verde. During August and September (Campaign 2) Cape Verde is subject to increased influence of North-easterly and Easterly air mass origins, with less influence from Westerly and North-westerly remote Atlantic air masses (Carpenter et al., 2010). This**

**shift coincides with more variable and enhanced glyoxal concentrations in easterly and north-easterly air masses (Fig. S5), which have had recent influence from the African coast and continent."**

*P16 line 13: ATom not AToM*

Thanks. This has been corrected.

*P20 line 14. "moderately elevated" is a better description here then significantly elevated*

We agree, and have updated this text accordingly:

**"glyoxal mixing ratios are moderately elevated compared to equivalent model runs with no monoterpene input.."**

*More explanation needed for Figure 8. The Figure 9 caption implies that these are diurnal means – is that correct? I find it intractable to compare the diurnal means described in Figure 9 with those from a scatter plot in Figure 8.*

Apologies for the confusion. This Figure 9 caption contained errors. It has been modified to:

**"Average observed (black) and modelled (red) glyoxal diurnal profiles for a) the first measurement period and 4 b) the second measurement period for the base model run. Shaded regions show the 1σ variability of the data. Note that the diurnal profiles include only data for which supporting measurements required to run the model are available and thus show small differences from the mean diurnal profiles displayed in Figure 6."**

*Figure 10 – Cumulative production and loss rates would be preferable in my view. The yellow legend doesn't show up well and see note on MCM names.*

The yellow colour has been changed to improve readability. While cumulative rates might be helpful in some ways, we argue that changing the plot in this way would make reading and comparison of individual loss and production rates (both with each other and with other studies) more difficult. We therefore have chosen to retain the plotting as originally presented.

*Figure S9 – label as a and b*

Labels have been added as suggested.

---

## Author Comment (AC2)

**Observations and modelling of glyoxal in the tropical Atlantic marine boundary layer – Response to Anonymous Referee #2**

We thank the reviewer for their positive and supportive comments on our manuscript. Below we respond to each specific comment in turn. Reviewer comments are shown in italic text, and our responses in normal text. Where we have added new text to the manuscript, this is summarised in bold text. Line numbers refer to the previously submitted original manuscript.

*Page 1, lines 13-14: The modeled glyoxal seems rather insensitive to aerosol effects, especially compared to the effects of acetaldehyde or sesquiterpenes. Later in the paper the authors say this (page 21, line33-34): "…the sensitivity of the modelled glyoxal to changes in the rate of aerosol uptake is not sufficient to reconcile the model with the observations." I would suggest changing the language in the abstract to be more consistent with the later text.*

This is a good point, and we have modified the abstract text, replacing this sentence with:

"**The model showed limited sensitivity to changes in deposition rates of model intermediates and the uptake of glyoxal onto aerosol compared with sensitivity to uncertainties in chemical precursors.**"

*Page 3 line 18-19: It would be good to also cite Lerot et al. 2021, who report glyoxal retrievals from TROPOMI, which like the other satellites also sees enhanced glyoxal over remote tropical oceans. The authors discuss several reasons why this might be the case.*

Thanks for drawing this to our attention. We have included a citation to this study, which we had previously missed in preparation of the manuscript:

"**The largest mixing ratios have been measured near coasts, and these measurements show less evidence for enhancement of glyoxal over remote tropical oceans as suggested by some satellite measurements (Vrekoussis et al., 2009; Lerot et al., 2010; 2021). Limited measurements of glyoxal in the free troposphere by airborne MAX-DOAS (Volkamer et al., 2015) may be consistent with the satellite-observed enhancements, although caution may be needed in interpretation of glyoxal satellite retrievals due to spectral inteferences (Lerot et al., 2021).**"

*Page 3, line 34-35: While 1.5e14 is the number from Lawson et al. (2015), it is a little confusing to compare a column measurement with an in situ measurement without also discussing the assumptions used to convert the in situ mixing ratio into a VCD. Stating that the satellite columns indicated higher levels of glyoxal than the in situ measurements would be fine.*

We agree with this point, and have modified the text to more explicitly state how the in situ and satellite data were compared by Lawson et al., (2015):
"**Assuming that all satellite-observed glyoxal resides in a well-mixed marine boundary layer of depth 850 m allowed a direct comparison of the in situ observations with GOME-2 vertical columns. This comparison suggested that the satellite observations exceeded the is situ observations by more than $1.5 \times 10^{14}$ molecule cm$^{-2}$ at both sites. However, this neglects the possibility of further glyoxal enhancements aloft, revealed by airborne measurements noted above.**"

*Page 14, line 8: Are there any measurement of aerosol composition, either at Cape Verde Atmospheric Observatory or from the ATom campaign, that could be used to better inform*

*the model?  Several of the references for the glyoxal uptake value (e.g. Volkamer 2007) are from studies in urban areas, where I would expect the aerosol to be mostly organic.  I'm not sure what effect the different ions in sea spray aerosol would have (e.g. Waxman et al. 2015), and a "real" number is better than a made up on, but it should be noted that an uptake coefficient for urban aerosol may not be representative of marine aerosol.*

While there is uncertainty in the aerosol uptake coefficient, which may be dependent on aerosol composition, we note that the model results are relatively insensitive to aerosol uptake (as also pointed out by the Reviewer above). We therefore do not feel that it would be worthwhile incorporating more information on aerosol composition in the model. Our sensitivity test to aerosol uptake (Fig. S10) also allows some measure of the magnitude of influence that may be expected, which is small. Nevertheless, we do agree with the reviewer that it is important to note that the aerosol uptake coefficient assumed is taken from data on continental aerosol, and may not be representative of marine aerosol. We have added a sentence to this effect (Page 14, line 8):
**"This value of $\gamma$ for glyoxal uptake is based on studies of continental aerosol, and so may not be representative for aerosol in the remote marine atmosphere. However, we find small sensitivity of simulated glyoxal in the model to the assumed value of $\gamma$."**

*The yellow font, and to a lesser extent the yellow traces, used in Figures 10, 12, and S10 is rather hard to read.  A darker shade of yellow for at least the legend would help.*

We apologise for the unclear yellow colour used in these figures. We have changed this to a more readable darker shade in these figures.

*I'm not sure what the ACP style guide says, but in Tables 1 and 2 I would use a lowercase "i" and "n" to abbreviate iso-butane and n-butane (and the other VOCs where this applies), to avoid confusion with nitrogen and iodine.*

Thanks for noting this. We have decided to change these prefixes to lower case to avoid confusion.

*Figures 5, 7, S1, and S2: Figure 5 uses day of month, while the other figures use Julian Day. It would be easier for the reader if a consistent date format, preferably that which was used in Figure 5, was used for all these figures.  Alternatively, dashed vertical lines on the Julian Day plots to indicate the first day of each month would work.*

All figures now use a consistent day of month time axis.